
# Charged quantum fields in AdS$_2$

**Dionysios Anninos$^{1\star}$, Diego M. Hofman$^{2\,\dagger}$ and Jorrit Kruthoff$^{2\,\S}$**

**1** Department of Mathematics, King's College London, the Strand, London WC2R 2LS, UK
**2** Institute for Theoretical Physics and $\Delta$ Institute for Theoretical Physics,
University of Amsterdam, Science Park 904, 1098 XH Amsterdam, The Netherlands

$\star$ dionysios.anninos@kcl.ac.uk, $\dagger$ d.m.hofman@uva.nl, $\S$ kruthoff@stanford.edu

## Abstract

We consider quantum field theory near the horizon of an extreme Kerr black hole. In this limit, the dynamics is well approximated by a tower of electrically charged fields propagating in an $SL(2, \mathbb{R})$ invariant AdS$_2$ geometry endowed with a constant, symmetry preserving background electric field. At large charge the fields oscillate near the AdS$_2$ boundary and no longer admit a standard Dirichlet treatment. From the Kerr black hole perspective, this phenomenon is related to the presence of an ergosphere. We discuss a definition for the quantum field theory whereby we 'UV' complete AdS$_2$ by appending an asymptotically two dimensional Minkowski region. This allows the construction of a novel observable for the flux-carrying modes that resembles the standard flat space $S$-matrix. We relate various features displayed by the highly charged particles to the principal series representations of $SL(2, \mathbb{R})$. These representations are unitary and also appear for massive quantum fields in dS$_2$. Both fermionic and bosonic fields are studied. We find that the free charged massless fermion is exactly solvable for general background, providing an interesting arena for the problem at hand.

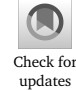

# 1 Introduction

The amount of angular momentum that can be acquired by a four-dimensional black hole is bounded by the square of its energy. In the limit where this bound is saturated something rather unusual occurs near the horizon. The geometry acquires an infinitely deep AdS$_2$ throat and the near horizon isometries are enhanced to an $SL(2,\mathbb{R}) \times U(1)$, which include the conformal group in one-dimension. Though it is clear how this happens at the level of general relativity, a microscopic understanding of this limit is surprisingly challenging. From general expectations based on the AdS/CFT correspondence one might suspect a simple answer: there exists some large $N$ quantum mechanical system with an $SL(2,\mathbb{R})$ symmetry dual to AdS$_2$. However, building a quantum mechanical system whose vacuum state preserves these symmetries is remarkably hard and infrared quantum effects tend to destroy the symmetries of classically $SL(2,\mathbb{R})$ invariant systems [1]. More recently, an investigation of systems with quenched disorder and a large $N$ limit have emerged as interesting toy models [2–8] for which the $SL(2,\mathbb{R})$ is preserved, at least to leading order in large $N$. However, most efforts have so far focused on the case with a two-dimensional bulk rather than the more realistic four-dimensional solutions of general relativity containing an AdS$_2$ factor.

    Part of the motivation for this work is to understand whether the lessons learned from the recent holographic models of AdS$_2$ holography can be applied to the extreme Kerr geometry [9].[1] There are several features that make this both interesting as well as challenging. The Kerr geometry exhibits an ergosphere which implies that there is no Killing vector which is everywhere timelike outside the horizon. The presence of an ergosphere is also a feature of the near horizon geometry for extreme Kerr. Relatedly, fields propagating in spacetimes with an ergosphere may exhibit superradiant phenomena [11, 26–28]. At the classical level this is a form of stimulated emission [29], by which an incoming wave comes out with a larger amplitude. At the quantum level this process becomes spontaneous and can be viewed as a rotational analogue of Hawking radiation [10, 27, 30–32]. Another motivation for our work

---

[1]Previous work on a holographic picture of the Kerr black hole and related theories enjoying $SL(2,\mathbb{R}) \times U(1)$ invariance includes [9–25].

comes from the fact that $SL(2, \mathbb{R})$ is also the symmetry group of dS$_2$. It is a curious feature that the symmetry group of de Sitter and anti-de Sitter are the same in two-dimensions. This may allow for a simpler bridge between our understanding of holography in AdS$_2$ and dS$_2$ [33,34] (see also [35–40] for related discussions).

Concretely, in this note we construct and analyse quantum fields near the horizon of an extreme Kerr black hole and assess several quantum states which behave interestingly under the symmetries at hand. Due to the ergosphere, this is a rather delicate question since we can no longer take for granted many of the usual features in quantum field theory. For instance, due to the presence of the ergosphere, the Hamiltonian is no longer guaranteed to have nice boundedness properties. That one finds regions of negative energy lies at the heart of the Penrose process [29], and we should not be surprised to encounter it in our analysis. Though quantum field theory in Kerr spacetimes is a rich and old problem [41,42], we will attempt to offer a new perspective for the extreme Kerr case by formulating the aforementioned issues in the language of $SL(2, \mathbb{R})$. In fact, the problem of either bosonic or fermionic fields in extreme Kerr directly maps to that of electrically charged quantum fields in an AdS$_2$ geometry endowed with a constant, symmetry preserving background electric field. When the charge is large enough, the modes can carry flux across the AdS$_2$ boundary. We connect the physics of such highly charged fields to the unitary principal series irreducible representation of $SL(2, \mathbb{R})$, thereby relating superradiance and representation theory in an interesting manner. Moreover, we find that the construction of an $SL(2, \mathbb{R})$ invariant state for bosonic fields is inevitably accompanied by an unbounded spectrum. As mentioned before, from the perspective of a spacetime with an ergosphere this is perhaps expected. However, somewhat surprisingly, we find that fermionic fields enjoy a bounded spectrum when the fermionic Hilbert space is constructed above a particular quantum state transforming in a highest weight representation of $SL(2, \mathbb{R})$. Finally, in allowing modes that carry flux across the AdS$_2$ boundary we are prompted to introduce a novel observable more akin to the flat-space S-matrix. We call this observable the $\mathcal{S}$-matrix and briefly assess some of its properties.

The paper is organised as follows. In section 2 we introduce the extreme Kerr near horizon geometry and its symmetries. In section 3 we analyse the free scalar field at the classical level. In section 4 we quantise this field. In section 5 we study the quantum fermionic field, and in section 6 we introduce a fermionic state with a bounded spectrum. Finally, in section 7 we offer some speculations about a holographic interpretation. Certain technical aspects and a discussion on dS$_2$ can be found in the appendix.

## 2 Geometry near the extreme Kerr horizon

The geometry we will consider is given by the near horizon region of an extreme Kerr black hole of mass $M$ and angular momentum $J = M^2$ (in units where the four-dimensional Newton constant is unity). In order to obtain it one must consider an infinite redshift of the asymptotically flat clock and a rescaling of the radial coordinate near the horizon of the full Kerr geometry. The resulting spacetime, itself a solution to Einstein's equations, takes the form [10]

$$ds^2 = \alpha(\theta)\left( ds^2_{\text{AdS}_2} + d\theta^2 \right) + \beta(\theta)(d\varphi + A)^2 \,, \tag{1}$$

with $\varphi \sim \varphi + 2\pi$ and $\theta \in (0, \pi)$. The explicit forms for $\alpha(\theta)$ and $\beta(\theta)$ are

$$\alpha(\theta) = J\left( 1 + \cos^2\theta \right) \,, \qquad \beta(\theta) = 4J\frac{\sin^2\theta}{1 + \cos^2\theta} \,. \tag{2}$$

For each fixed polar angle $\theta$, the geometry (1) is a type of Hopf fibration of an $S^1$ fibre space over an AdS$_2$ base-space. The one-form $A$ depends explicitly on the base-space coordinates.

The fibre-space is compact given that the coordinate $\varphi$ parametrises a circle. We can express the AdS$_2$ base space in several coordinate systems. In Poincaré coordinates, the base space and one-form are:

$$ds^2_{\text{AdS}_2} = \frac{-dt^2 + dz^2}{z^2} \,, \qquad A = \frac{dt}{z} \,, \tag{3}$$

with $t \in \mathbb{R}$ , and $z \in (0, \infty)$ . In these coordinates, the original black hole horizon lies at $z = \infty$. We will also be interested in the global chart of AdS$_2$. To obtain this, we consider a (complexified) coordinate transformation:

$$t \pm z = \pm i e^{i(\tau \mp \rho)} \tag{4}$$

accompanied by a simple (complexified) $U(1)$ gauge transformation, to obtain:

$$ds^2 = \frac{-d\tau^2 + d\rho^2}{\cos^2 \rho} \,, \qquad A = \tan \rho \, d\tau \,. \tag{5}$$

Here $\rho \in (-\pi/2, \pi/2)$ such that there are two asymptotic boundaries. From the perspective of the extreme Kerr horizon, one of the two boundaries lives in the interior of the horizon. Finally, we can also consider black hole coordinates, which can be obtained from the global chart by taking $\rho \to ir + \pi/2$ and $\tau \to -iT$, such that:

$$ds^2 = \frac{-dT^2 + dr^2}{\sinh^2 r} \,, \qquad A = \coth r \, dT \,. \tag{6}$$

The horizon lives at $r \to \infty$, whereas the AdS$_2$ boundary now resides at $r = 0$. The electric field is non-vanishing at the horizon.

The Killing symmetries of the geometry are given by $SL(2, \mathbb{R}) \times U(1)$. The $U(1)$ corresponds to constant shifts in the azimuthal angle $\varphi$. We denote the generators of the Lie algebra $\mathfrak{sl}(2, \mathbb{R})$ by $\{\mathfrak{H}, \mathfrak{K}, \mathfrak{D}\}$. We have:

$$[\mathfrak{H}, \mathfrak{D}] = i\mathfrak{H} \,, \qquad [\mathfrak{D}, \mathfrak{K}] = i\mathfrak{K} \,, \qquad [\mathfrak{H}, \mathfrak{K}] = 2i\mathfrak{D} \,. \tag{7}$$

In the Poincaré coordinate system these generators are represented by the following Killing vectors:

$$\xi_{\mathfrak{H}} = i \, \partial_t \,, \qquad \xi_{\mathfrak{D}} = i \, (t \partial_t + z \partial_z) \,, \qquad \xi_{\mathfrak{K}} = i \left( \frac{z^2}{2} + \frac{t^2}{2} \right) \partial_t + i \left( tz \partial_z - z \partial_\varphi \right) \,. \tag{8}$$

Asymptotically, at small and constant $z$, these Killing symmetries act purely on $t$ as the standard generators of smooth conformal maps of the real line to itself. In global coordinates (5), the algebra (7) is generated by the Killing vectors

$$\xi_{\mathfrak{H}} = i(1 - \cos\tau \sin\rho)\partial_\tau - i \sin\tau \cos\rho \, \partial_\rho - i \cos\tau \cos\rho \, \partial_\varphi \,, \tag{9}$$

$$\xi_{\mathfrak{D}} = -i \sin\tau \sin\rho \, \partial_\tau + i \cos\tau \cos\rho \, \partial_\rho - i \sin\tau \cos\rho \, \partial_\varphi \,, \tag{10}$$

$$\xi_{\mathfrak{K}} = i(1 + \cos\tau \sin\rho)\partial_\tau + i \sin\tau \cos\rho \, \partial_\rho + i \cos\tau \cos\rho \, \partial_\varphi \,. \tag{11}$$

The compact generator $\mathfrak{R} = (\mathfrak{K} + \mathfrak{H})/2$ acts on fields by a $\tau$-translation. When we consider the universal cover of $SL(2, \mathbb{R})$ we must decompactify $\tau$.

It is important to note that *none* of the Killing symmetries are globally timelike. Moreover, none are everywhere timelike outside the horizon. For instance:

$$\xi_{\mathfrak{H}} \cdot \xi_{\mathfrak{H}} = A^2 \left( \alpha(\theta) - \beta(\theta) \right) \tag{12}$$

becomes spacelike for $\theta \in (0.82..., 2.32...)$. The absence of an everywhere timelike Killing vector in the near horizon geometry indicates that the near horizon geometry exhibits physical

properties akin to an ergosphere. Thus, phenomena associated to an ergosphere such as the Penrose process [29] and its quantum analogue [30, 31] will be present in the near horizon region. Nevertheless, constant $t$, $\tau$ or $T$ surfaces are spacelike, and as such we can use either $t$ or $\tau$ as clocks. Even so, given that $\partial_t$, $\partial_\tau$, and $\partial_T$ are spacelike for certain values of $\theta$ means that the associated charges need not be bounded from below. This is the reason we can 'extract' energy from a rotating black hole by entering the ergosphere [29]. In the black hole coordinates (6), the Killing vector $\chi = i(\partial_T - \Omega_H \partial_\varphi)$ generates the horizon, where $\Omega_H$ is the angular velocity of the horizon. Sufficiently close to the $\text{AdS}_2$ boundary, $\chi$ becomes spacelike.

Given the presence of an $SL(2, \mathbb{R}) \times U(1)$ symmetry group, we have the opportunity to consider quantum fields in an extreme Kerr black hole background from a different, more group theoretical, perspective. To this end, we conclude with a brief discussion of the representation theory of $SL(2, \mathbb{R})$.

## 2.1 Unitary representations of $SL(2, \mathbb{R})$

Given our interest in quantum fields on an $\text{AdS}_2$ background, it will be useful to briefly review some facts about the unitary irreducible representations of $SL(2, \mathbb{R})$. These have been classified in [43, 44] (for a recent discussion see [45]). $SL(2, \mathbb{R})$ is the group of real valued two-by-two matrices with unit determinant. The centre of the group is $Z = \{\pm \mathbb{I}_2\}$. Given that $SL(2, \mathbb{R})$ is non-compact, its non-trivial unitary irreducible representations are all infinite dimensional.

As mentioned earlier, we label the $\mathfrak{sl}(2, \mathbb{R})$ generators by $\{\mathfrak{H}, \mathfrak{D}, \mathfrak{K}\}$ obeying the algebra (7). A basis for the Lie algebra is given by the traceless real two-by-two matrices. One of the $\mathfrak{sl}(2, \mathbb{R})$ generators, namely $\mathfrak{R} = (\mathfrak{H} + \mathfrak{K})/2$, generates the compact $U(1)$ subgroup of $SL(2, \mathbb{R})$. It is also convenient to define the non-compact generator $\mathfrak{S} = (\mathfrak{K} - \mathfrak{H})/2$. The quadratic Casimir is given by:

$$\mathfrak{C}_2 = \mathfrak{D}^2 - \frac{1}{2}(\mathfrak{H}\mathfrak{K} + \mathfrak{K}\mathfrak{H}) = \mathfrak{D}^2 + \mathfrak{S}^2 - \mathfrak{R}^2 \equiv \lambda(1 - \lambda). \tag{13}$$

Here $\lambda \in \mathbb{C}$ is a convenient label which will be used to denote the various irreducible representations. It is related to the scaling properties of a given state. At this point it is worth pointing out the following: given that $\mathfrak{so}(1, 2) = \mathfrak{so}(2, 1) \cong \mathfrak{sl}(2, \mathbb{R})$, one cannot distinguish between a Euclidean and Lorentzian one-dimensional conformal algebra. This is in sharp contrast to higher dimensions where there is a clear difference between the Euclidean and Lorentzian conformal algebra, i.e. $\mathfrak{so}(3, 2) \neq \mathfrak{so}(4, 1)$.

The Hilbert space of a theory with an $SL(2, \mathbb{R})$ symmetry can be partitioned into the unitary irreducible representations. In addition to $\lambda$, the states $|m\rangle_\lambda$ can be labelled by the eigenvalue $m \in \mathbb{Z}$ of the compact subgroup $\mathfrak{R}$. Reality of $\mathfrak{C}_2$ enforces that $\lambda$ is either real or of the form $\lambda = 1/2 + is$ with $s \in \mathbb{R}$. Representations with $\lambda = 1/2 + is$ and $m \in \mathbb{Z}$ are known as principal series representations. There are two distinct non-trivial unitary irreducible representations for real $\lambda$. The complementary series, which have $\lambda \in (0, 1)$ and $m \in \mathbb{Z}$, and the two discrete series with $\lambda \in \mathbb{Z}^+$ and either $m = \lambda, \lambda + 1, \ldots$ or $m = -\lambda, -\lambda - 1, \ldots$ The discrete series of $SL(2, \mathbb{R})$ often referred to as highest/lowest weight representations. We note that the principal series and complementary series have a positive Casimir, whereas the discrete series have a non-positive Casimir. We will mostly be interested in the universal cover of $SL(2, \mathbb{R})$, the eigenvalues of $\mathfrak{R}$ are no longer required to be integer valued. Finally, it is worth mentioning that the principal series representation is not a unitary irreducible representation of the Virasoro algebra unless the central extension vanishes [46].[2]

It is convenient to introduce generators $L_\pm$ that act as raising and lowering operators on the eigenstates $|m\rangle_\lambda$ of $\mathfrak{R}$. They are constructed as linear combinations of the generators

---

[2]We would like to acknowledge useful discussions with T. Anous on this point.

$\{\mathfrak{S}, \mathfrak{D}, \mathfrak{R}\}$

$$L_\pm = \frac{1}{2}(\mathfrak{K} - \mathfrak{H}) \mp i\mathfrak{D} = \mathfrak{S} \mp i\mathfrak{D} , \tag{14}$$

and obey the algebra

$$[\mathfrak{R}, L_\pm] = \mp L_\pm , \qquad [L_+, L_-] = 2\mathfrak{R} . \tag{15}$$

The $L_\pm$ act on $|m\rangle_\lambda$ as

$$L_\pm |m\rangle_\lambda = (\pm(\lambda - 1) - m)|m \pm 1\rangle_\lambda . \tag{16}$$

For both the principal and complementary series, there is no state $|m\rangle_\lambda$ that is annihilated by either $L_+$ or $L_-$. Therefore, the the tower of states extends indefinitely across both positive and negative $m$ for such irreducible representations.

It is also instructive to discuss the transformation properties of conformal operators $\mathcal{O}_\lambda(m)$ of weight $\lambda$. We work in a eigenbasis of the compact generator $\mathfrak{R}$. The transformation properties of conformal operators are given by

$$[L_\pm, \mathcal{O}_\lambda(m)] = (\pm(\lambda - 1) - m)\mathcal{O}_\lambda(m \pm 1) , \qquad [\mathfrak{R}, \mathcal{O}_\lambda(m)] = -m\,\mathcal{O}_\lambda(m) , \tag{17}$$

from which we build local operators

$$\mathcal{O}_\lambda(\tau) = \sum_{m \in \mathbb{Z}} e^{-im\tau} \mathcal{O}_\lambda(m) , \qquad \tau \sim \tau + 2\pi . \tag{18}$$

When the operators are conformal *primary* operators in the highest/lowest weight representations they commute with one of the generators of $\mathfrak{sl}(2, \mathbb{R})$. For the principal series representations this is not the case.

The generators $\{\mathfrak{S}, \mathfrak{D}, \mathfrak{R}\}$ are hermitian operators, from which it follows that $L_\pm = L_\mp^\dagger$. With this notion of hermitian conjugation, we define the bra states $_\lambda\langle m|$, conjugate to $|m\rangle_\lambda$. We can also define conjugate operators. The simplest way of doing that is to take the hermitian conjugate (the one with respect to which $L_+ = L_-^\dagger$) of the transformation properties of $\mathcal{O}_\lambda(m)$. If $\lambda$ is real, this gives rise to an operator $\mathcal{O}_\lambda^\dagger(m)$ that has the same weight as $\mathcal{O}_\lambda(m)$. When $\lambda$ is complex this is not the case as the weight is conjugated as well. However, it is still possible to define a different kind of conjugate operator that does have the same weight. In particular, taking the hermitian conjugate of (17) gives

$$[L_\mp, \mathcal{O}_\lambda^\dagger(m)] = (\mp(\lambda^* - 1) + m)\mathcal{O}_\lambda^\dagger(m \pm 1) . \tag{19}$$

Since the operator $\mathcal{O}_\lambda$ is complex, let us define $\mathcal{P}_\lambda(m)$ as $\mathcal{O}_\lambda^\dagger(m) \equiv e^{i\delta_m}\mathcal{P}_\lambda(-m)$. We now demand that $\mathcal{P}_\lambda(m)$ has the same weight as $\mathcal{O}_\lambda(m)$. This fixes the phases $\delta_m$ to satisfy the following recursion relation:

$$e^{i\delta_{m\pm 1}} = \frac{\pm\lambda^* + m}{\pm\lambda + m} e^{i\delta_m}. \tag{20}$$

Notice that these phases are non-trivial when $\lambda$ is complex. Plugging in $\lambda = 1/2 + is$, we find

$$e^{i\delta_m} = \frac{\Gamma(1/2 - is + m)}{\Gamma(1/2 + is + m)} . \tag{21}$$

The local operator $\mathcal{P}_\lambda(\tau)$ is then

$$\mathcal{P}_\lambda(\tau) = \sum_{m \in \mathbb{Z}} e^{-i\delta_{-m}}\mathcal{O}_\lambda^\dagger(-m)e^{-im\tau} . \tag{22}$$

This means that $\mathcal{P}_\lambda(\tau)$ is the shadow operator of $\mathcal{O}_\lambda^\dagger(\tau)$. To see this notice that we can write a product of Fourier modes as a convolution and that

$$\sum_{m \in \mathbb{Z}} e^{-i\delta_m}e^{im\tau} = c_\lambda\left(\sin^{-2}\tau/2\right)^{1/2+is} , \qquad c_\lambda = \frac{\Gamma(2\lambda)\cos(\pi\lambda)}{2^{2\lambda-1}} . \tag{23}$$

We then have

$$\mathcal{P}_\lambda(\tau) = \int_{S^1} \frac{d\tau'}{2\pi} c_\lambda \left( \sin^{-2} \frac{\tau - \tau'}{2} \right)^{1/2+is} \mathcal{O}_\lambda^\dagger(\tau') , \qquad (24)$$

from which it is clear that $\mathcal{P}_\lambda(\tau)$ has weight $\lambda = 1/2 + is$. Below, we will derive that the kernel in (24) is indeed a conformal invariant two-point function built from (complex) operators in the principal series representation.

Given an $SL(2,\mathbb{R})$ invariant state $|0\rangle$ that is annihilated by all three generators of the Lie algebra we can consider two- and three-point functions of the $\mathcal{O}_\lambda(\tau)$. These are fixed by the symmetries. For instance, let us consider the two point function in frequency space, $c_{m,m'}^{\lambda\lambda'} \equiv \langle 0|\mathcal{O}_\lambda(m)\mathcal{O}_{\lambda'}(m')|0\rangle$. We can use the $SL(2,\mathbb{R})$ invariance of the vacuum and (17) to deduce that

$$0 = \langle 0|\mathfrak{R}\mathcal{O}_\lambda(m)\mathcal{O}_{\lambda'}(m')|0\rangle = -(m+m')c_{m,m'}^{\lambda\lambda'} . \qquad (25)$$

This allows us to write $c_{m,-m}^{\lambda\lambda'} \equiv c_m^{\lambda\lambda'}$. Inserting $L_\pm$ instead, we find

$$c_{m\pm1}^{\lambda\lambda'}(\pm(\lambda-1)-m) + c_m^{\lambda\lambda'}(\pm\lambda'+m) = 0 . \qquad (26)$$

For generic $\lambda$ and $\lambda'$ there are no solutions to (26). However, by picking $\lambda = \lambda'$ or $\lambda = 1 - \lambda'$ there is a solution given by

$$c_m^{\lambda\lambda'} = \alpha_\lambda \delta_{\lambda\lambda'} \frac{\Gamma(\lambda+m)}{\Gamma(1-\lambda+m)} + \beta_\lambda \delta_{\lambda,1-\lambda'} , \qquad (27)$$

up to irrelevant $m$-independent constants $\alpha_\lambda$ and $\beta_\lambda$. This leads to

$$\langle 0|\mathcal{O}_\lambda(\tau)\mathcal{O}_{\lambda'}(0)|0\rangle = \gamma_\lambda \delta_{\lambda\lambda'} \left( \sin^2 \tau/2 \right)^{-\lambda} + \zeta_\lambda \delta_{\lambda,1-\lambda'} \delta(\tau) . \qquad (28)$$

The coefficients $\gamma_\lambda$ and $\zeta_\lambda$ are the usual normalisations of the two-point function. For the principal series representation $\lambda = 1/2 + is$ and hence the ultra-local piece in (28) is present whenever $\lambda' = \lambda^*$ instead of just when $\lambda = \lambda' = 1/2$. We also see now that the kernel in (24) is indeed a conformal two-point function of two complex operators in the principal series with weight $\lambda = 1/2 + is$, such that (24) can be viewed as a type of shadow transform. We will see examples of both types of correlators in section 5 and appendix B. Correlators for $\lambda = \lambda' = 1/2$ containing both terms in (28) where found in [9]. One could also have derived these correlators by staying in position space and using the following differential operator representation of the $\mathfrak{sl}(2,\mathbb{R})$ generators,

$$R = i\partial_\tau, \quad L_\pm = ie^{\pm i\tau}\partial_\tau \pm (\lambda-1)e^{\pm i\tau} , \qquad (29)$$

to constrain their functional form.

A simple example of a quantum system realising the $SL(2,\mathbb{R})$ symmetries is a quantum particle moving on the Poincaré disk. Another system realising the $SL(2,\mathbb{R})$ symmetries is quantum field theory in a fixed (A)dS$_2$ background. We discuss the dS$_2$ case in appendix B. In what follows, we will concern ourselves with the possible role of the principal series representations in the context of the extreme Kerr geometry. As suggested by the geometry itself, this problem is intimately connected to quantum field theory of a charged particle propagating in a fixed AdS$_2$ geometry in the presence of a background electric field [32, 47–49].

# 3  Classical scalar field

In this section, we consider a massless scalar field propagating in the fixed background (1) at the classical level. Upon Fourier expanding

$$\Phi(t,z,\theta,\varphi) = \sum_{l,m} \int_{\mathbb{R}} \frac{d\omega}{2\pi} e^{-it\omega+im\varphi} \mathcal{Y}_{lm}(\theta) \psi_{lm\omega}(z) . \tag{30}$$

Remarkably, the wave-equation separates [50]. One piece is an equation for the spheroidal harmonics, $\mathcal{Y}_{lm}(\theta)$, which is a generalised version of the spherical harmonic equation. One finds a discretum of eigenvalues $j_{lm}$ bounded from below. The other piece governs the radial part, and takes the form of the wave-equation of a charged, massive particle propagating in $AdS_2$ in the presence of a background gauge field [10]:

$$\left( -(\mathcal{D}_t^A)^2 + \partial_z^2 - \frac{j_{lm}-m^2}{z^2} \right) e^{-i\omega t} \psi_{lm\omega}(z) = 0 . \tag{31}$$

Here $\mathcal{D}_\mu^A \equiv (\partial_\mu - imA_\mu)$ is a $U(1)$ gauge-covariant derivative in a fixed $AdS_2$ background of curvature $-1/J$. The background gauge field is given by $A_t = 1/z$ and the electric charge is $m$. The mass squared of the particle is given in terms of the eigenvalue $j_{lm}$ of the spheroidal harmonics. We are thus led to analyze the problem of a quantum field in a fixed $AdS_2$ background endowed with a constant electric field.

## 3.1  Hamiltonian

To simplify notation, we will examine the following action:

$$S = \int dt\,dz \left( |\mathcal{D}_t^A\Phi|^2 - |\partial_z\Phi|^2 - \frac{\mu^2}{z^2}|\Phi|^2 \right) , \tag{32}$$

where $\Phi(t,z)$ is a complex, charged scalar field of mass $\mu^2$ and charge $q$. The background fields are:

$$ds^2 = \frac{-dt^2+dz^2}{z^2} , \qquad A_t dt = \frac{dt}{z} , \tag{33}$$

with $z \in (0,\infty)$ and $t \in \mathbb{R}$. The symmetry of our background is given by $SL(2,\mathbb{R})\times U(1)$.[3] The $U(1)$ acts as a constant phase rotation of the complex scalar field. We note that the background gauge field $A_t$ transforms by a gauge transformation under the $SL(2,\mathbb{R})$, such that the field-strength $dA$ is indeed $SL(2,R) \times U(1)$ invariant. The dilatations and time translations are manifest in the problem. The special conformal transformations act as follows:

$$\Phi(t,z) \to \mathcal{L}_{\zeta_\mathfrak{K}}\Phi(t,z) + qz\,\Phi(t,z) , \qquad \zeta_\mathfrak{K}^\mu\partial_\mu = i\left( \frac{z^2+t^2}{2}\partial_t + tz\partial_z \right) . \tag{34}$$

The conserved current associated with the $\mathfrak{u}(1)$ generator is:

$$\mathcal{J}_\mu = i\left( \Phi^* \mathcal{D}_\mu^A\Phi - \Phi(\mathcal{D}_\mu^A\Phi)^* \right) . \tag{35}$$

The conserved currents associated with the $\mathfrak{sl}(2,\mathbb{R})$ generators are given by [51]:

$$\mathcal{J}_\mu^\xi = \xi^\nu\left( T_{\nu\mu} - A_\nu\mathcal{J}_\mu \right) + \alpha_\xi\mathcal{J}_\mu , \tag{36}$$

---

[3]It is interesting that in $AdS_2$, a background electric field preserves the full conformal group. From the perspective of a putative dual conformal quantum mechanics, we would view this as a marginal deformation, at least to leading order in the large $N$ limit [9]. An electric field in higher dimensional AdS breaks the conformal symmetries, and the solution has a charged black brane in the interior.

where $T_{\mu\nu}$ is the following (non-conserved) tensor:

$$T_{\mu\nu} = (\mathcal{D}^A_\mu \Phi)^* \mathcal{D}^A_\nu \Phi + (\mathcal{D}^A_\nu \Phi)^* \mathcal{D}^A_\mu \Phi - g_{\mu\nu}\big((\mathcal{D}^A_\sigma \Phi)^* \mathcal{D}^\sigma_A \Phi - \mu^2 |\Phi|^2\big) \tag{37}$$

and $\alpha_\xi$ is defined through $\mathcal{L}_\xi A = d\alpha_\xi$ to ensure conservation of (36).

From the action (32) we obtain the conjugate field momentum:

$$\Pi = \left(\partial_t + \frac{iq}{z}\right)\Phi^* , \tag{38}$$

and consequently we can construct the generator of $t$-translations:

$$H_t = \int_0^\infty dz \left[\left(\Pi - \frac{iq}{z}\Phi^*\right)\left(\Pi^* + \frac{iq}{z}\Phi\right) + |\partial_z \Phi|^2 + \frac{(\mu^2 - q^2)}{z^2}|\Phi|^2\right] . \tag{39}$$

Viewing $H_t$ as the classical Hamiltonian, we observe that the classical vacuum depends on the ratio $\mathcal{R} \equiv \mu^2/q^2$. For $\mathcal{R} > 1$ the classical vacuum sits at the origin, whereas for $\mathcal{R} < 1$ there is a runaway direction where we take $\Phi$ constant, $\Pi = iq\Phi^*/z$ and $|\Phi|^2 \to \infty$. Thus, for $\mathcal{R} < 1$, the classical Hamiltonian is unbounded from below. From the perspective of the Kerr background, where $\partial_t$ is not an everywhere timelike Killing vector outside the horizon, the unboundedness of the Hamiltonian for $\mathcal{R} < 1$ reflects the presence of an ergoregion.

## 3.2 Classical solutions in Poincaré AdS$_2$

We will now analyse the behaviour of the classical solutions for both $\mathcal{R} > 1$ and $\mathcal{R} < 1$. Explicitly, the equation of motion governing a charged massive scalar field, $\Phi(t,z)$ is

$$\left[-\left(\partial_t - \frac{iq}{z}\right)^2 + \partial_z^2\right]\Phi = \frac{\mu^2}{z^2}\Phi . \tag{40}$$

We can exploit the time translation invariance and expand $\Phi$ in a Fourier expansion:

$$\Phi(t,z) = \int_{\mathbb{R}} \frac{d\omega}{2\pi} e^{-i\omega t} \Phi_\omega(z) . \tag{41}$$

The restriction to $\omega \in \mathbb{R}$ amounts to restricting the solutions to ones which are not exponentially growing in the future or past. Since $\Phi(t,z)$ is complex, there are no reality conditions on $\Phi_\omega(z)$. The equation governing the Fourier modes becomes an ordinary differential equation:

$$\left(\frac{d^2}{dz^2} + \left(\omega + \frac{q}{z}\right)^2 - \frac{\mu^2}{z^2}\right)\Phi_\omega(z) = 0 . \tag{42}$$

For each value of $\omega$, there are two complex solutions to the above equation.

We will write down explicit solutions for this equation shortly, but first we will assess their behaviour in certain limits. At large $z$ the equation reduces to that for plane waves with solutions $\Phi_\omega(z) \approx \alpha^\pm_\omega e^{\pm iz\omega}$ with $\alpha^\pm_\omega \in \mathbb{C}$. Thus, the fields behave as plane waves near the Poincaré horizon. For small values of $z$, the equation is approximated by:

$$\left(\frac{d^2}{dz^2} + \frac{q^2 - \mu^2}{z^2}\right)\Phi_\omega(z) = 0 . \tag{43}$$

The scaling properties of the above equation lead to solutions of the form $\Phi_\omega(z) \approx \beta^\pm_\omega z^{\lambda_\pm}$, with $\beta^\pm_\omega \in \mathbb{C}$. One readily finds:

$$\lambda(1-\lambda) = q^2 - \mu^2 \quad \longleftrightarrow \quad \lambda_\pm = \frac{1}{2} \pm \sqrt{\frac{1}{4} + \mu^2 - q^2} . \tag{44}$$

Given a solution to (42) with $\omega$ and $q$, it follows that there will also be a solution upon exchanging $(\omega, q) \rightarrow -(\omega, q)$. Also, given a solution to (42) with $\lambda$, there is also a solution with $\lambda \rightarrow (1 - \lambda)$. It is important to note that for small enough values of $\mathcal{R}$, the parameter $\lambda$ becomes complex and the solutions exhibit oscillatory behaviour near the AdS$_2$ boundary.

The full solution of (40) is given in terms of the two Whittaker functions:

$$\Phi_\omega(z) = \alpha_\omega M_{-iq, \nu}(2iz\omega) + \beta_\omega W_{-iq, \nu}(2iz\omega), \qquad \nu = \sqrt{\frac{1}{4} + \mu^2 - q^2}. \tag{45}$$

We note that $\alpha_\omega$ and $\beta_\omega$ are complex. One can check that the asymptotic properties of these solutions match our previous approximations. The parameter $\nu$ is either real and positive or an imaginary number $\nu = is$ with $s$ real. Given a solution, we can act with the discrete transformations to generate another. For instance, with $\nu = is$ we have that $W_{iq, -is}(-2i\omega z) = \left(W_{-iq, is}(2i\omega z)\right)^*$ and $M_{iq, -is}(-2i\omega z) = \left(M_{-iq, is}(2i\omega z)\right)^*$. It is useful to note the following asymptotic expansions:

$$\lim_{x \rightarrow 0} M_{-iq, \nu}(ix) = (ix)^{1/2+\nu} + \dots, \tag{46}$$

$$\lim_{x \rightarrow \infty} M_{-iq, \nu}(ix) = e^{i\pi\nu/2 - \pi q/2} \Gamma(1 + 2\nu) \left(\frac{x^{iq} e^{ix/2 - i\pi\nu/2}}{\Gamma(1/2 + \nu + iq)} + \frac{i x^{-iq} e^{-ix/2 + i\pi\nu/2}}{\Gamma(1/2 + \nu - iq)}\right), \tag{47}$$

as well as:

$$\lim_{x \rightarrow 0} W_{-iq, \nu}(ix) = \left(\frac{\Gamma(2\nu)(ix)^{1/2-\nu}}{\Gamma\left(iq + \nu + \frac{1}{2}\right)} + \frac{\Gamma(-2\nu)(ix)^{1/2+\nu}}{\Gamma\left(iq - \nu + \frac{1}{2}\right)}\right) + \dots, \tag{48}$$

$$\lim_{x \rightarrow \infty} W_{-iq, \nu}(ix) = e^{\pi q/2} e^{-ix/2} x^{-iq} + \dots. \tag{49}$$

We now discuss some properties for $\nu$ real and pure imaginary.

**Case I: $\nu \in \mathbb{R}^+$**

When $\nu \in \mathbb{R}^+$, it follows from (46) that it is possible to impose Dirichlet boundary conditions near the AdS$_2$ boundary at $z \rightarrow 0$. In doing so, we select two of the four complex solutions. These are $M_{-iq, \nu}(2iz\omega)$ and its complex conjugate. Near the Poincaré horizon, these solutions become oscillatory and the solution becomes a linear combination of incoming and outgoing waves. Inspecting (47) reveals that the two waves have the equal amplitude, such that the net flux though the Poincaré horizon vanishes. Thus, for real $\nu$, the problem is qualitatively similar to that of neutral fields in AdS$_2$.

**Case II: $\nu = is$ with $s \in \mathbb{R}$**

When $\nu = is$ with $s \in \mathbb{R}$ the solution changes qualitatively. Near the AdS$_2$ boundary the solutions are of the form $\Phi_\omega(z) \sim (\omega z)^{1/2 \pm is}$ and hence become oscillatory. Thus, the waves carry flux across the AdS$_2$ boundary. There is no longer a sense in which we can naturally impose Dirichlet boundary conditions at $z = 0$. Moreover, inspecting (47) reveals that a purely left/right moving excitation near the AdS$_2$ boundary becomes a linear combination of incoming and outgoing waves near the Poincaré horizon.

Since net flux is generally carried through the Poincaré horizon, it is instructive to also consider the theory in global coordinates (5).

### 3.3 Classical solutions in global AdS$_2$ with $\nu = is$

In global coordinates, the metric is given in (5) and the equation of motion for our charged particle becomes:

$$\left( \frac{\partial^2}{\partial \rho^2} + (\omega + q \tan \rho)^2 - \frac{\mu^2}{\cos^2 \rho} \right) \Phi_\omega(\rho) = 0 , \tag{50}$$

where we have used the Fourier decomposition

$$\Phi(\tau, \rho) = \int_{\mathbb{R}} \frac{d\omega}{2\pi} e^{-i\omega\tau} \Phi_\omega(\rho) . \tag{51}$$

Using the Killing vectors (9)-(11), and replacing $-i\partial_\varphi$ with $q$, the wave equation is equivalent to the quadratic Casimir such that $\Phi_\omega(\rho)$ satisfies the eigenvalue problem

$$\mathfrak{C}_2 \Phi_\omega(\rho) = \left( \frac{1}{4} + q^2 \right) \Phi_\omega(\rho) . \tag{52}$$

There are two complex solutions to the above equation, one of which is given by

$$\Phi_\omega^{(1)}(\rho) = z^a (1-z)^b \frac{{}_2F_1(1 - \lambda - iq, \lambda - iq, 1 + 2a, z)}{\Gamma(1 - \omega - iq)} , \tag{53}$$

and the other, which we denote by $\Phi_\omega^{(2)}(\rho)$, is obtained by sending $(\omega, q)$ to $(-\omega, -q)$ in (53). Here we have defined $z = (1 + i\tan\rho)/2$, $a = -(\omega + iq)/2$, $b = (\omega - iq)/2$ and $\lambda = 1/2 + is$. The functions $\Phi_\omega^{(1,2)}(\rho)$ are smooth functions of $\rho$. Under certain linear combinations of the Killing vectors in (9)-(11) given by

$$\mathcal{L}_0 = \frac{1}{2}\left( \xi_{\mathfrak{K}} + \xi_{\mathfrak{H}} \right), \qquad \mathcal{L}_\pm = \frac{1}{2}\left( \xi_{\mathfrak{K}} - \xi_{\mathfrak{H}} \right) \mp i\xi_{\mathfrak{D}} , \tag{54}$$

the solutions acquire a ladder-type structure

$$\mathcal{L}_+ \Phi_\omega^{(1)} = i\Phi_{\omega+1}^{(1)} , \qquad\qquad \mathcal{L}_- \Phi_\omega^{(1)} = -i(\lambda - \omega)(1 - \lambda - \omega)\Phi_{\omega-1}^{(1)} \tag{55}$$

$$\mathcal{L}_+ \Phi_\omega^{(2)} = i(\lambda + \omega)(1 - \lambda + \omega)\Phi_{\omega+1}^{(2)} , \qquad\qquad \mathcal{L}_- \Phi_\omega^{(2)} = -i\Phi_{\omega-1}^{(2)} , \tag{56}$$

and $\mathcal{L}_0 \Phi_{\lambda,\omega}^{(1,2)} = -\omega \Phi_{\lambda,\omega}^{(1,2)}$. For complex $\lambda$ and real $\omega$ the tower of solutions built by acting with $\mathcal{L}_\pm$ will never terminate.

Given that $\lambda \in \mathbb{C}$ the solutions $\Phi_{\lambda,\omega}^{(1,2)}$ are oscillatory near the global AdS$_2$ boundaries. For instance, $\lim_{\rho \to \pi/2} \Phi_{\lambda,\omega}^{(1)}(\rho) \sim (\pi/2 - \rho)^\lambda + \gamma_\omega(\pi/2 - \rho)^{1-\lambda}$ for some $\omega$-dependent coefficient $\gamma_\omega$. Using these two solutions we can thus construct a linear combination that is purely left or right moving near the right boundary. In particular,

$$\Phi_{\omega,R}^{(+)}(\rho) = \Phi_\omega^{(1)}(\rho) + \alpha(\omega)\Phi_\omega^{(2)}(\rho) , \tag{57}$$

with

$$\alpha(\omega) = -e^{\pi(q - i\omega)} \frac{\Gamma(\lambda + iq)\Gamma(\lambda + \omega)}{\Gamma(\lambda - iq)\Gamma(\lambda - \omega)} , \tag{58}$$

behaves as $\mathcal{N}_\omega(\rho - \pi/2)^\lambda$ in the limit $\rho \to \pi/2$. The subscript $R$ refers to the right boundary. Near the left boundary $\Phi_{\omega,R}^{(+)}(\rho)$ is a linear combination of left and right movers.[4] That an

---

[4]This is most easily seen from the following identity for hypergeometric functions

$$
{}_2F_1(\alpha, \beta, \gamma, z) = \frac{\Gamma(\gamma)\Gamma(\gamma - \alpha - \beta)}{\Gamma(\gamma - \alpha)\Gamma(\gamma - \beta)} {}_2F_1(\alpha, \beta, \alpha + \beta - \gamma + 1, 1 - z)
$$
$$
+ (1-z)^{\gamma - \alpha - \beta} z^{1-\gamma} \frac{\Gamma(\gamma)\Gamma(\alpha + \beta - \gamma)}{\Gamma(\alpha)\Gamma(\beta)} {}_2F_1(1 - \beta, 1 - \alpha, \gamma - \alpha - \beta + 1, 1 - z) .
$$

incoming wave from the past can turn into a linear combination of left and right moving outgoing waves in the future can be viewed as the classical analogue of particle production. The mode $\Phi^{(+)}_{\omega,R}$ again has the ladder structure under the action of (54),

$$\mathcal{L}_+ \Phi^{(+)}_{\omega,R} = i\Phi^{(+)}_{\omega+1,R}\,, \quad \mathcal{L}_- \Phi^{(+)}_{\omega,R} = -i(\lambda-\omega)(1-\lambda-\omega)\Phi^{(+)}_{\omega-1,R}\,, \quad \mathcal{L}_0 \Phi^{(+)}_{\omega,R} = -\omega\Phi^{(+)}_{\omega,R}\,. \quad (59)$$

A similar story holds for the other three possibilities for having a single type of oscillation near one of the boundaries of $AdS_2$.

# 4 Quantum scalar field

In this section we consider the quantisation of a charged scalar field in a fixed $AdS_2$ background endowed with a constant electric field. We first consider the case of real $\nu$, which admits a standard $SL(2,\mathbb{R})$ preserving quantisation procedure akin to the case of neutral quantum fields in AdS. We then discuss the case with $\nu = is$ and motivate the necessity for a different approach.

## 4.1 Poincaré $AdS_2$ construction with $\nu \in \mathbb{R}^+$

Let us begin by considering the standard Fock space construction for the case $\nu \in \mathbb{R}^+$, where things are more transparent.

Imposing standard Dirichlet conditions at the $AdS_2$ boundary gives us two independent mode functions, namely $\Phi_\omega(z) = M_{-iq,\nu}(2iz\omega)$ and its complex conjugate. We express the local quantum field operator in terms of a collection of mode operators as:

$$\hat{\Phi}(t,z) = \int_{\omega>0} \frac{d\omega}{2\pi}\left(\frac{e^{-i\omega t}}{\sqrt{\omega}}\hat{a}_\omega \Phi_\omega(z) + \frac{e^{i\omega t}}{\sqrt{\omega}}\hat{b}^\dagger_\omega \Phi_{-\omega}(z)\right)\,. \quad (60)$$

The field momentum $\hat{\Pi}(t,z)$ can be similarly defined:

$$\hat{\Pi}(t,z) = i\int_{\omega>0} \frac{d\omega}{2\pi}\left(\frac{e^{i\omega t}}{\sqrt{\omega}}\left(\omega+\frac{q}{z}\right)\hat{a}^\dagger_\omega (\Phi_\omega(z))^* - \frac{e^{-i\omega t}}{\sqrt{\omega}}\left(\omega-\frac{q}{z}\right)\hat{b}_\omega (\Phi_{-\omega}(z))^*\right)\,. \quad (61)$$

At equal times, the operators must satisfy the Heisenberg algebra $[\hat{\Phi}(t,z),\hat{\Pi}(t,z')] = i\delta(z-z')$. Assuming that $\hat{a}_\omega$ and $\hat{b}_\omega$ commute, this imposes

$$\int_{\omega_{1,2}>0} \frac{d\omega_1}{2\pi}\frac{d\omega_2}{2\pi}\left(e^{-it(\omega_1-\omega_2)}\frac{(\omega_2+q/z)}{\sqrt{\omega_1\omega_2}}\Phi_{\omega_1}(z)(\Phi_{\omega_2}(z'))^*[\hat{a}_{\omega_1},\hat{a}^\dagger_{\omega_2}]\right.$$
$$\left. + e^{it(\omega_1-\omega_2)}\frac{(\omega_2-q/z)}{\sqrt{\omega_1\omega_2}}\Phi_{-\omega_1}(z)(\Phi_{-\omega_2}(z'))^*[\hat{b}_{\omega_1},\hat{b}^\dagger_{\omega_2}]\right) = \delta(z-z')\,. \quad (62)$$

The time-independence of the right hand side of (62) imposes that the commutators are proportional to $\delta(\omega_1-\omega_2)$. Let us denote the algebra satisfied by $\hat{a}_\omega$ and $\hat{b}_\omega$ as

$$[\hat{a}_\omega,\hat{a}^\dagger_{\omega'}] = \alpha(\omega)\delta(\omega-\omega')\,, \qquad [\hat{b}_\omega,\hat{b}^\dagger_{\omega'}] = \beta(\omega)\delta(\omega-\omega')\,. \quad (63)$$

Finally, the scaling symmetry $z \to \lambda z$ imposes that $\alpha(\omega)$ and $\beta(\omega)$ are in fact constant. The operator algebra (63) requires that the modes functions obey

$$\int_{\omega>0} \frac{d\omega}{2\pi}\left[\alpha\left(1+\frac{q}{\omega z}\right)\Phi_\omega(z)\left(\Phi_\omega(z')\right)^* + \beta\left(1-\frac{q}{\omega z}\right)\Phi_{-\omega}(z)\left(\Phi_{-\omega}(z')\right)^*\right] = \delta(z-z')\,. \quad (64)$$

Notice that despite appearances, the integrand has no pole at $\omega = 0$ due to the behaviour of the modes near there.

Though we have not managed to prove (64), we believe it to be true provided that $\alpha = \beta$. When $z \neq z'$, the integrand is an oscillatory function suggesting cancellations leading to a vanishing integral. Moreover, by scaling $\omega \to \lambda \omega$, we can fix the value of $z$. For instance, we can push it to large values, where the modes are highly oscillatory as seen in the asymptotic expression (47). For both $z$ and $z'$ large, the terms quadratic in $\Phi$ give terms that go as $\omega^{\pm 2iq} e^{\pm i\omega(z+z')}$. Once we integrate over $\omega$ such terms will give non-trivial functions of $z + z'$ that would not be present on the left hand side of (64). In order for these pieces to vanish, we have to pick $\alpha = \beta$. As a result, we can combine the two terms on the right hand side of (64) into an integral over $\omega \in \mathbb{R}$. Finally, we can fix $\alpha$ by requiring a unit coefficient in front of $\delta(z-z')$. This yields,

$$\alpha = \frac{e^{\pi q}|\Gamma(1/2 - iq + \nu)|^2}{4\,\Gamma(1+2\nu)^2} \ . \tag{65}$$

Due to the absence of flux leaking through the Poincaré horizon when $\nu \in \mathbb{R}$ we can also obtain a completeness type of relation. Indeed, it follows from the wave-equation (42) that

$$\int_0^\infty dz \left( \omega + \omega' + \frac{2q}{z} \right) \Phi_\omega(z) (\Phi_{\omega'}(z))^* = \mathcal{N}(\omega) \delta(\omega - \omega') \ , \tag{66}$$

with

$$\mathcal{N}(\omega) = 4\pi\omega \frac{e^{\pi q}\Gamma(1+2\nu)^2}{|\Gamma(1/2 + iq + \nu)|^2} \ . \tag{67}$$

Using (66) we can integrate the operators $\hat{\Phi}(t,z)$ and $\hat{\Pi}(t,z)$ against $z$ for constant $t$ to retrieve $\hat{a}_\omega$ and $\hat{b}_\omega$. For instance

$$\hat{a}_\omega = -i\sqrt{\omega} \int_0^\infty dz\, \Phi_\omega(z) \left( \hat{\Pi}(0,z) + i \left( \omega + \frac{q}{z} \right) \hat{\Phi}^\dagger(0,z) \right) \ . \tag{68}$$

Alternatively, we can isolate the modes $\hat{a}_\omega$ and $\hat{b}_\omega$ by taking linear combinations of $\hat{\Phi}$ and $\hat{\Pi}$. For instance:

$$\hat{a}_\omega = \lim_{z \to \infty} \sqrt{\omega}\, e^{\pm iz\omega} \int_{\mathbb{R}} dt\, e^{i\omega t} \left( \hat{\Phi}(t,z) \pm \frac{i}{\omega} \partial_z \hat{\Phi}(t,z) \right) \ . \tag{69}$$

Our theory allows for an $SL(2,\mathbb{R}) \times U(1)$ invariant state $|0\rangle$ defined by $\hat{a}_\omega |0\rangle = \hat{b}_\omega |0\rangle = 0$. In this state, we can calculate two-point functions of our boundary operators, $\hat{\mathcal{O}}(t) = \lim_{z \to 0} z^{-1/2 - \nu} \hat{\Phi}(t,z)$. For example:

$$\langle 0| \hat{\mathcal{O}}(t_1) \hat{\mathcal{O}}^\dagger(t_2) |0\rangle = \frac{c_\Delta}{(t_1 - t_2)^{2\Delta}} \ , \tag{70}$$

with

$$c_\Delta = e^{-i\pi\Delta} \frac{2^{2\Delta}\Gamma(2\Delta)}{2\pi} \alpha \ , \tag{71}$$

and $\Delta = 1/2 + \nu$. That the correlation functions are $SL(2,\mathbb{R}) \times U(1)$ covariant supports the fact that the background electric field indeed preserves the full symmetry group.

## 4.2 The $\mathcal{S}$-matrix as an alternative approach for $\nu = is$

Having discussed the more standard case with $\nu \in \mathbb{R}^+$ we now turn to $\nu = is$ with $s \in \mathbb{R}$. As we already observed when constructing classical solutions in section 3.3, the fields now carry flux across the AdS$_2$ boundary. The phenomenon is intimately related to the fact that

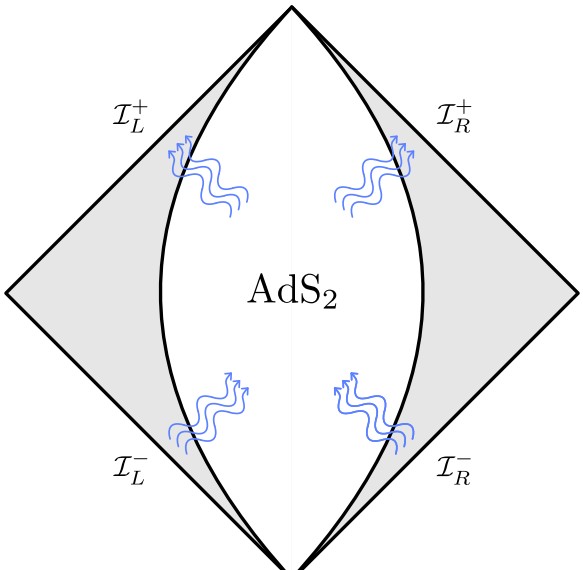

Figure 1: The geometric setup for the $\mathcal{S}$-matrix. The shaded regions are the asymptotically flat regions with vanishing electric field. On their asymptotic boundaries $\mathcal{I}_{R/L}^{\pm}$, we define Hilbert spaces $\mathcal{H}_{R/L}^{\pm}$. The region in the middle is an AdS$_2$ region with constant electric field and flux (blue) is allowed through its boundary.

the scaling dimension $\lambda$ in (44) becomes complex for $\nu = is$. This is not the standard situation one encounters upon quantising fields in AdS. We are thus confronted with the necessity of building the appropriate observables for $\nu = is$.

We will consider the problem in the global AdS$_2$ chart since any seemingly natural condition on the fields will allow flux through the Poincaré horizon. We will consider a novel type of observable, more akin to the flat space $S$-matrix rather than a collection of correlation functions along the timelike boundaries of global AdS$_2$. The reason for doing so is that we would like to permit states that carry flux in and out of the AdS$_2$ boundary, which is qualitatively similar to what happens to fields in asymptotically flat space. The flat space S-matrix most naturally lives on a null asymptotic boundary where we can build an asymptotic Fock space of states. The challenge in building an S-matrix in global AdS$_2$ is the absence of such an asymptotic null boundary. We will circumvent this by appending an auxiliary flat region to global AdS$_2$, which as we shall see is a sensible procedure specifically in two spacetime dimensions. In turn, this will allow us to define a object akin the the flat space $S$-matrix – an object we refer to as the AdS$_2$ $\mathcal{S}$-matrix.

To sharpen our considerations, consider a charged scalar field of charge $q$ propagating in a fixed metric and background gauge field of the more general form[5]

$$ds^2 = e^{2\sigma(\rho)}(-d\tau^2 + d\rho^2), \qquad A(\rho) = qf(\rho)d\tau, \tag{72}$$

where the coordinate $\rho$ spans the whole real line. We can embed global AdS$_2$ into an asymptotically flat spacetime by requiring $f(\rho)$ and $\sigma(\rho)$ to have the certain features. The interpolating region occurs at two constant-$\rho$ surfaces $\rho = \pm\rho_c$ with $\rho_c \equiv (\pi/2 - \epsilon)$ and $\epsilon \ll 1$ a small positive number. For $|\rho| < \rho_c$, $f(\rho) = \tan\rho$ and $e^{2\sigma(\rho)} = \cos^{-2}\rho$, such that we have a global AdS$_2$ geometry with a constant electric field. For $|\rho| > \rho_c$, we take $f(\rho) = \tan(\rho_c)\operatorname{sgn}(\rho)$ and $e^{2\sigma(\rho_c)} = \cos^{-2}\rho_c$ such that the geometry becomes two-dimensional flat space with vanishing electric field. In the limit $\epsilon \to 0$, the interpolating region becomes parametrically small.

---

[5]For example, arising from time-translation invariant solutions to general dilaton-Maxwell theories.

Though the extension of global $AdS_2$ into asymptotically flat space seems innocent enough, our construction is only possible in two-dimensions.[6]

Each asymptotically flat region contains a past and future null boundary $\mathcal{I}^{\pm}_{R/L}$, with $R/L$ denoting the right/left boundary, as shown in figure 1. At each of these, we define a Hilbert space $\mathcal{H}^{\pm}_{R/L}$ of asymptotic states. The setup allows us to consider the overlap between the past and future Hilbert spaces:

$$S = \langle \mathcal{H}^+_R \otimes \mathcal{H}^+_L | \mathcal{H}^-_R \otimes \mathcal{H}^-_L \rangle \,, \tag{73}$$

thus defining an $S$-matrix satisfying the usual unitarity condition $S^{\dagger} S = 1$.

The explicit construction of the $S$-matrix is somewhat cumbersome for charged scalars. For massless charged fermions the technical aspects of the problem become significantly simpler. Moreover, the consideration of charged fermions propagating in $AdS_2$ with a background electric field reveals several novel phenomena as compared to the charged scalar. Consequently, we now proceed to consider the case of massless charged fermions.

# 5 Quantum fermionic field

The propagation of a neutral four-component Dirac fermion on a fixed Kerr black hole background is separable into a time-dependent part, angular part and radial part [52–55]. The angular part is governed by a spin one-half generalisation of the spheroidal harmonics and can be expressed as a spin-weighted spherical harmonics. In the near horizon limit of the extreme Kerr geometry, the radial wave equation reduces to that of a charged two-component Dirac fermion moving in an $AdS_2$ background endowed with a constant electric field. The two components of this fermion completely determine the four-component spinor in the massive case. In the massless case the two chiralities decouple and one is left with two two-component spinors, which have opposite frequency and charge as seen from the near horizon perspective [56]. The separation constant plays the role of a time-reversal breaking mass term.

We are thus prompted to study a Dirac fermion in two-dimensional anti-de Sitter spacetime with constant electric field. For simplicity, we focus on a massless, electrically charged fermion. We moreover assume there is no time-reversal symetry breaking mass. The massive case is relegated to appendix A.

## 5.1 Integrability in general background and symmetries

We begin by considering the propagation of a massless, charged fermion $\Psi$ on the general background (72). Our conventions for the gamma matrices, which satisfy the usual Clifford algebra $\{\gamma^a, \gamma^b\} = 2\eta^{ab}$, are taken to be $\gamma^0 = i\sigma_1$ and $\gamma^1 = \sigma_3$. The Lorentzian action governing our massless fermions is then given by

$$S = -i \int d^2x \sqrt{-g}\, \bar{\Psi}\gamma^{\mu}\mathcal{D}^A_{\mu}\Psi \,, \tag{74}$$

with $\bar{\Psi} = \Psi^{\dagger}\gamma^0$. For the general background (72) the spin connection is given by $\omega^{01} = \partial_{\rho}\sigma(\rho)\, d\tau$.

In the absence of a background gauge field, the symmetries of a two-dimensional free massless fermion are given by two copies of the infinite Virasoro algebra. In the presence of a background gauge field the symmetry group will differ. To see this it is convenient to work with null coordinates $x_{\pm} \equiv \rho \pm \tau$. An infinitesimal conformal Killing vector $\zeta \equiv \zeta^-(x_-)\partial_- + \zeta^+(x_+)\partial_+$

---

[6]In higher dimensions, the analogue of our construction would extend global AdS into an asymptotically Einstein static universe.

of (72) will multiply the metric by a conformal factor, but change the gauge field $A$ by a Lie derivative:

$$A \rightarrow A + \mathcal{L}_\zeta A \,. \tag{75}$$

For the action to remain invariant, we must further consider a local phase transformation of the fermionic fields $\delta\Psi = i\alpha\Psi$, where $\alpha(x_+, x_-)$ must satisfy:

$$d\alpha = \mathcal{L}_\zeta A \,. \tag{76}$$

Given an arbitrary background gauge field $A$, the condition (76) cannot be solved for arbitrary conformal Killing vector. Taking an exterior derivative of (76), we get the following integrability condition:

$$\mathcal{L}_\zeta F = 0 \,. \tag{77}$$

The metric under consideration depends only on $\rho = (x_+ + x_-)/2$ with a constant background electric field, and the above equation becomes

$$F_{+-}(\nabla_+ \zeta^+ + \nabla_- \zeta^-) = 0 \,. \tag{78}$$

This equation is satisfied for arbitrary $\zeta$ whenever $F_{+-} = 0$. On the other hand, if we want to find solutions for an arbitrary gauge potential with non-vanishing $F_{+-}$, we require that $(\nabla_+ \zeta^+ + \nabla_- \zeta^-)$ vanishes – i.e. that $\zeta$ is a Killing vector. For those $\zeta$ that are symmetries, we can employ the Noether procedure, and obtain the conserved currents

$$\mathcal{J}_\zeta^\mu = \zeta^\nu (T_\nu{}^\mu - A_\nu \mathcal{J}^\mu) + \alpha_\zeta \mathcal{J}^\mu, \tag{79}$$

where $\mathcal{J}^\mu = \bar{\Psi}\gamma^\mu\Psi$ is the conserved $U(1)$ current and $T_{\mu\nu}$ the stress tensor, which, due to the non-zero electric field, satisfies $\nabla_\mu T^{\mu\nu} = \mathcal{J}_\alpha F^{\alpha\nu}$, with $F^{\alpha\nu}$ the fieldstrength associated to $A_\mu$.

We now show that equations of motion stemming from the action (74) are integrable for an arbitrary background (72). To this end, it is convenient to define $\Psi = (1 + i\sigma_1)\chi/\sqrt{2}$ such that the equations of motion become:

$$\left(\partial_\tau \mp \partial_\rho \mp Q_\pm(\rho)\right)\chi_\pm(\tau, \rho) = 0 \,, \qquad Q_\pm(\rho) \equiv \frac{1}{2}\partial_\rho \sigma(\rho) \mp iA_\tau \,. \tag{80}$$

Further defining

$$\chi_\pm(\tau, \rho) = \exp\left(-\int_{\rho_0}^{\rho} Q_\pm(\rho')\,d\rho'\right) g^\pm(\tau, \rho) \,, \tag{81}$$

we see that $g^\pm(\tau, \rho)$ satisfies the same equations of motion as a neutral fermion in flat space. Even though the equations of motion are integrable, it is interesting that the system does not exhibit an infinite collection of conserved quantities. In fact, by viewing the equations of motion (80) as those of a neutral fermion propagating in a curved background with complexified Weyl factor $\tilde{\sigma}(\rho) = \sigma(\rho) \mp 2iq \int^\rho A_\tau(\rho)$, there is a sense in which the system exhibits an infinite collection of 'complexified symmetries'. However, the would be conserved charges implied by these complexified symmetries do not obey standard reality conditions.

Although we have concentrated on Weyl factors and gauge fields that depend purely on $\rho$, the free massless and charged fermion is in fact solvable for the most general two dimensional metric (in conformal gauge) and background gauge field $A_\mu(\tau, \rho)$. We leave the details of this to future work.

## 5.2  Solutions in global AdS$_2$ and quantisation

The global AdS$_2$ metric and background gauge field is a special case of the general background (72), namely $e^{2\sigma(\rho)} = \cos^{-2}\rho$ and $f(\rho) = \tan\rho$ with $\rho \in (-\pi/2, \pi/2)$. Applying the discussion of the previous subsection to this background, one finds that the problem is mapped to a neutral, massless fermion on a flat strip of spatial length $\pi$. The symmetries of the system are now $SL(2,\mathbb{R}) \times U(1)$. Explicitly we have:

$$\zeta_0 = i\partial_\tau \,, \qquad \zeta_\pm(x_+, x_-) = \pm\left(e^{\pm ix_+}\Phi_+ + e^{\mp ix_-}\Phi_-\right), \tag{82}$$

$$\alpha_{\zeta_\pm}(x_+, x_-) = -\frac{iq}{2}\left(e^{\pm ix_+} + e^{\mp ix_-}\right), \tag{83}$$

and $\alpha_{\zeta_0} = 0$.[7] The general solution for the flat strip is given by $g^\pm(\tau, \rho) = f^\pm(x_\pm)$ where $f^\pm(z)$ are arbitrary complex functions. Hence, using (81) we find:

$$\Psi(\tau, \rho) = (\cos\rho)^{1/2+iq} f^+(x_+)\eta_+ + (\cos\rho)^{1/2-iq} f^-(x_-)\eta_- \,, \tag{84}$$

where

$$\eta_+ = \frac{1}{\sqrt{2}}\begin{pmatrix} 1 \\ i \end{pmatrix}, \qquad \eta_- = \frac{1}{\sqrt{2}}\begin{pmatrix} i \\ 1 \end{pmatrix}. \tag{85}$$

Due to the complex exponent in the cosine, the fields carry non-vanishing flux across the AdS$_2$ boundaries. In this case, the phenomenon occurs for any non-vanishing value of $q$.

A null slice in global AdS$_2$ heading from one boundary at time $\tau$ to the other connects the points $(-\pi/2, \tau)$ to $(\pi/2, \tau + \pi)$. Consequently, along the null slice starting at $\tau = -\pi/2$ the function $f^+(x_+)$ takes values in the range $x_+ \in (-\pi, \pi)$ (and similarly for $f^-(x_-)$). We proceed with a null quantisation of these waves. Since left and right movers decouple, it is natural to quantise them separately. A complete basis for functions on an interval of length $2\pi$ is expressed in a Fourier expansion

$$f_\pm(x_\pm) = \sum_{n\in\mathbb{Z}} e^{inx_\pm} f_n^\pm \,. \tag{86}$$

Here $f_n^\pm$ are complex operators. Upon quantisation we promote $f_n^\pm$ to operators which satisfy the anti-commutation relations:

$$\{\hat{f}_m^\sigma, (\hat{f}_n^{\sigma'})^\dagger\} = \delta^{\sigma,\sigma'}\delta_{m,n} \,, \tag{87}$$

with $\sigma, \sigma' \in \{\pm\}$ The fermionic field operator can now be expressed as:

$$\hat{\Psi}^\pm(\tau, \rho) = (\cos\rho)^{1/2\pm iq}\,\eta_\pm \sum_{n\in\mathbb{Z}} e^{inx_\pm} \hat{f}_n^\pm \,, \tag{88}$$

with a similar expression for $(\hat{\Psi}^\pm)^\dagger$.

*Closing the open quantum system*

We remind the reader that the problem on the strip is incomplete. At the classical level, this is due to the fact that we are dealing with a Cauchy incomplete problem since flux can leave or enter the timelike boundary of the strip. At the quantum level, we have an open quantum system.

One approach to deal with this is to impose Dirichlet boundary conditions (or perhaps some more general Robin type condition), thus relating the left and right moving sectors. The

---

[7]The constant part of $\alpha_\zeta$ is ambiguous, but can be fixed by demanding orthogonality between the currents in (79) and the $U(1)$ current.

approach we take is different. As shown in figure 1, we embed the strip into an asymptotically two-dimensional flat spacetime thereby completing the problem into an $\mathcal{S}$-matrix framework – hence restoring (perturbative) unitarity. One can then explicitly introduce a cutoff $\varepsilon > 0$, making the strip have spatial length $(\pi - 2\varepsilon)$ and having $\sigma(\rho)$ and $f(\rho)$ converging to constant values outside the strip so that the outside is asymptotically flat space. This has the mild effect of correcting the $\mathfrak{sl}(2,\mathbb{R})$ generators at order $\varepsilon$ and so the full symmetry is broken only by order $\varepsilon$ terms.[8] Moreover, in our approach the left and right moving sectors of the massless fermion remain decoupled at the free level.

Given the exact solution (81) for arbitrary background geometry, a particularly explicit treatment is encountered for charged massless fermions. The integrability can be used to check that the $\mathcal{S}$-matrix is unity for free massless fermions propagating in an arbitrary background for which $\sigma(\rho)$ and $f(\rho)$ are odd functions of $\rho$.

### 5.3 Generators of $\mathfrak{sl}(2,\mathbb{R}) \times \mathfrak{u}(1)$ and the principal series

It is instructive to represent the $\mathfrak{sl}(2,\mathbb{R})$ generators through the $\hat{f}^{\pm}$ operators. This can be done by expressing the currents $\mathcal{J}^{\mu}$ in (79) in terms of $\hat{f}^{\pm}$ and then integrating the $+/-$ component over $x_{\pm}$, respectively, to get a set of integrated operators $\{\hat{L}_0, \hat{L}_{\pm}\}$ that form an $\mathfrak{sl}(2,\mathbb{R})$ algebra. Notice that upon quantisation, one either chooses a null slice along $x_+$ or $x_-$ direction to define a Hilbert space. Hence, only the $+$ component or $-$ component of the current contributes to $\{\hat{L}_0, \hat{L}_{\pm}\}$. Let us focus on quantising on slices parametrised by $x_+$, i.e. left-movers. Then

$$\hat{L}_0 = -\sum_{n \in \mathbb{Z}} n (\hat{f}_n^+)^{\dagger} \hat{f}_n^+ , \qquad \hat{L}_{\pm} = \sum_{n \in \mathbb{Z}} (\mp \lambda - n)(\hat{f}_{n \pm 1}^+)^{\dagger} \hat{f}_n^+ , \tag{89}$$

with $\lambda = 1/2 + iq$. The Hermiticity conditions are given by $\hat{L}_+^{\dagger} = \hat{L}_-$ and $\hat{L}_0^{\dagger} = \hat{L}_0$. Furthermore, the $\mathfrak{u}(1)$ generator $\hat{Q}$ in the left-moving sector reads

$$\hat{Q} = -i \sum_{n \in \mathbb{Z}} (\hat{f}_n^+)^{\dagger} \hat{f}_n^+ . \tag{90}$$

In writing (89) and (90), we have chosen a particular normal-ordering prescription. This takes operator valued functions $\mathcal{F}(\hat{f}_n^+, (\hat{f}_n^+)^{\dagger})$ and pushes the $\hat{f}_n^+$ to the right. The operators $\{\hat{L}_0, \hat{L}_{\pm}\}$, satisfy the $\mathfrak{sl}(2,\mathbb{R})$ algebra:

$$[\hat{L}_0, \hat{L}_{\pm}] = \mp \hat{L}_{\pm} , \qquad [\hat{L}_+, \hat{L}_-] = 2\hat{L}_0 , \tag{91}$$

and commute with $\hat{Q}$. Given (89), we can define an $SL(2,\mathbb{R}) \times U(1)$ invariant state $|\Omega\rangle$. We define:

$$\hat{f}_n^+ |\Omega\rangle = 0, \qquad n \in \mathbb{Z} . \tag{92}$$

Acting with $(f_n^+)^{\dagger}$ on $|\Omega\rangle$ we can further construct a discrete family of states:

$$|n\rangle = (\hat{f}_n^+)^{\dagger} |\Omega\rangle , \quad n \in \mathbb{Z} . \tag{93}$$

These states are all normalisable eigenstates of $\hat{L}_0$ with eigenvalue $-n$. Furthermore,

$$\hat{L}_{\pm} |n\rangle = (\pm(-1/2 - iq) - n)|n \pm 1\rangle . \tag{94}$$

---

[8]Perhaps an alternative view of our setup is as a boundary field theory type problem, where we are adding additional degrees of freedom/interactions near the AdS$_2$ boundary. In field theories with a boundary, bulk parameters such as the electric charge or mass of the field are not expected to be renormalised by boundary interactions [57]. From this perspective, we do not expect the physics of charged particles within AdS$_2$ to be significantly modified by appending the asymptotically flat region.

From the above expression we see that the family $|n\rangle$ does not furnish a highest-weight representation of $SL(2,\mathbb{R})$ — there is no state $|n\rangle$ annihilated by $\hat{L}_\pm$. Rather, the tower furnishes a principal series representation with weight $1/2-iq$ (comparing with our definition (16)) and Casimir $\mathfrak{C}_2 = 1/4+q^2$.

We can also consider the conformal properties of the field operator $\hat{\Psi}^\pm$. A straightforward calculation reveals

$$[\hat{L}_a, \hat{\Psi}^\sigma(\tau,\rho)] = -\mathcal{L}_{\zeta_a}\hat{\Psi}^\sigma(\tau,\rho)\,, \qquad a \in \{0,\pm 1\}\,, \quad \sigma \in \{\pm\}\,, \tag{95}$$

where the Lie derivative $\mathcal{L}_\zeta$ acts on spinors as

$$\mathcal{L}_\zeta \hat{\Psi}^\sigma(\tau,\rho) = -i\alpha_\zeta \hat{\Psi}^\sigma(\tau,\rho) + \zeta^\alpha \nabla_\alpha \hat{\Psi}^\sigma(\tau,\rho) + \frac{1}{8}(\nabla_\alpha \zeta^\beta - \nabla_\beta \zeta^\alpha)\gamma_\alpha\gamma_\beta\hat{\Psi}^\sigma(\tau,\rho)\,, \tag{96}$$

where $\zeta$ is an AdS$_2$ Killing vector, and we have suppressed the spinor indices. The indices $\alpha, \beta$ run over the global AdS$_2$ coordinates $\{\tau,\rho\}$. From $\hat{\Psi}^\sigma(\tau,\rho)$ we can construct the boundary operators at $\rho = \pi/2$ such as

$$\hat{\mathcal{O}}_\lambda(\tau) = \lim_{\rho\to\pi/2}(\cos\rho)^{-1/2-iq}\hat{\Psi}^+(\tau,\rho) = \eta_+ \sum_{n\in\mathbb{Z}} i^n e^{in\tau}\hat{f}_n^+\,, \tag{97}$$

and a similar boundary operator at $\rho \to -\pi/2$. Acting with $\hat{\mathcal{O}}_\lambda^\dagger(\tau)$ on $|\Omega\rangle$. We can compute the two-point function of $\hat{\mathcal{O}}_\lambda^\dagger(\tau)$ in $|\Omega\rangle$. The simplest two-point function is:

$$\langle\Omega|\hat{\mathcal{O}}_\lambda(\tau_1)\hat{\mathcal{O}}_\lambda^\dagger(\tau_2)|\Omega\rangle = \sum_{n\in\mathbb{Z}} e^{in(\tau_1-\tau_2)} = 2\pi\delta(\tau_1-\tau_2)\,, \tag{98}$$

where we contracted the spinor indices. To obtain a more interesting two-point function, we use the construction of the shadow operator $\hat{\mathcal{P}}_\lambda(\tau)$ discussed in section 2.1. This can be viewed as the boundary value of a bulk field $\hat{\mathcal{P}}_\lambda(\tau,\rho)$

$$\hat{\mathcal{P}}_\lambda(\tau,\rho) = (\cos\rho)^{1/2+iq}\overline{\eta}_+ \sum_{n\in\mathbb{Z}} e^{-i\delta_n}e^{-in\tau}(f_n^+)^\dagger\,, \tag{99}$$

in the sense that in the limit[9]

$$\hat{\mathcal{P}}_\lambda(\tau) = \lim_{\rho\to\pi/2}(\cos\rho)^{-1/2-iq}\hat{\mathcal{P}}_\lambda(\tau,\rho)\,. \tag{100}$$

Here $e^{i\delta_n}$ is given by (21). Like $\hat{\mathcal{O}}_\lambda(\tau)$, the boundary operator $\hat{\mathcal{P}}_\lambda(\tau)$ transforms as a principal series operator with weight $\lambda = 1/2+iq$. We can now consider the two-point function

$$\langle\Omega|\hat{\mathcal{O}}_{\lambda,\alpha}(\tau_1)\hat{\mathcal{P}}_{\lambda,\beta}(\tau_2)|\Omega\rangle = \eta_{\alpha,+}\overline{\eta}_{\beta,+} \sum_{n\in\mathbb{Z}} e^{2i\delta_n}e^{in(\tau_1-\tau_2)}, \tag{101}$$

with $\alpha, \beta$ spinor indices. The sum can be performed explicitly and yields

$$\langle\Omega|\hat{\mathcal{O}}_{\lambda,\alpha}(\tau_1)\hat{\mathcal{P}}_{\lambda,\beta}(\tau_2)|\Omega\rangle = \eta_{\alpha,+}\overline{\eta}_{\beta,+}\frac{\Gamma(2\lambda)\cos(\pi\lambda)}{2^{2\lambda-1}}\left(\sin^{-2}\frac{\tau_1-\tau_2}{2}\right)^{1/2+iq}\,, \tag{102}$$

as expected from symmetry considerations.

---

[9]Notice that this is really $\hat{\mathcal{P}}_\lambda(-\tau)$, but to avoid clutter we just denote it as $\hat{\mathcal{P}}_\lambda(\tau)$.

*Dirac sea and the ergosphere*

From the perspective of the standard fermionic ground state construction, the state $|\Omega\rangle$ corresponds to an unfilled Dirac sea (see also [11] for an interesting connection between super-radiance and Fermi-Dirac statistics). The spectrum of $\hat{L}_0$ for the states created on top of $|\Omega\rangle$ is unbounded. This is the price to pay if we are to preserve the $SL(2,\mathbb{R})$ symmetry in a theory furnishing the principal series representation with $\hat{L}_0$ as the Hamiltonian. If our system originates from a quantum field theory near the horizon of an extreme Kerr geometry, $\tau$ is not the clock related to an observer outside the horizon. In such a context, it is no longer clear whether we should interpret $\hat{L}_0$ as a Hamiltonian. Moreover, given that the neither the full Kerr geometry nor its near horizon region enjoys an everywhere timelike Killing vector outside the horizon, positivity of energy in the near horizon region may no longer be the correct physical requirement. One might go even further, and argue that what should be considered is the full system containing both the black hole as well as the quantum field. To justify the approximation of a quantum field in a fixed black hole background, the mass and angular momentum of the background black hole should be parametrically large. From this perspective, again, it is not clear that one should impose a positive energy condition on the spectrum of the quantum field alone. Nevertheless, we should still replace the absence of boundedness with another physical condition. Perhaps unitarity of the perturbative $\mathcal{S}$-matrix is sufficient. We will return to this question in future work.

To ameliorate the situation we now proceed to build a state, which we refer to as the $|\Upsilon\rangle$ vacuum, that breaks the $SL(2,\mathbb{R})$ but preserves boundedness. Even though $|\Upsilon\rangle$ breaks $SL(2,\mathbb{R})$, we shall see that it sits at the bottom of a lowest weight representation.

# 6 The fermionic $|\Upsilon\rangle$ vacuum

In this section we discuss a state $|\Upsilon\rangle$ leading to an $\hat{L}_0$ spectrum bounded from below. The state can be viewed as a Dirac sea filled with all negative energy particles above $|\Omega\rangle$.

## 6.1 Definition of $|\Upsilon\rangle$

We define $|\Upsilon\rangle$ to be annihilated by $(\hat{f}_n^+)^\dagger$ for $n > 0$, and $\hat{f}_n^+$ for $n < 0$. Excitations above $|\Upsilon\rangle$ all have non-negative $\hat{L}_0$ eigenvalues The operators $\{L_0, \hat{L}_\pm\}$ introduced in (89) are somewhat singular from the perspective of the Hilbert space built on top of $|\Upsilon\rangle$. This can be improved by modifying the normal ordering prescription, giving rise to a new set of $\mathfrak{sl}(2,\mathbb{R})$ generators

$$\hat{L}_0 = \frac{1}{2}\lambda(1-\lambda) + \sum_{n\geq 0} n\left(\hat{f}_n^+(\hat{f}_n^+)^\dagger + (\hat{f}_{-n}^+)^\dagger \hat{f}_{-n}^+\right), \tag{103}$$

$$\hat{L}_+ = \sum_{n\geq 0}\left[(\lambda+n)\hat{f}_n^+(\hat{f}_{n+1}^+)^\dagger + (1-\lambda+n)(\hat{f}_{-n}^+)^\dagger f_{-n-1}^+\right], \tag{104}$$

$$\hat{L}_- = \sum_{n\geq 0}\left[(1-\lambda+n)\hat{f}_{n+1}^+(\hat{f}_n^+)^\dagger + (\lambda+n)(\hat{f}_{-n-1}^+)^\dagger \hat{f}_{-n}^+\right], \tag{105}$$

with $\lambda = 1/2 + iq$. The commutation relations are again those of $\mathfrak{sl}(2,\mathbb{R})$:

$$[\hat{L}_0, \hat{L}_\pm] = \mp\hat{L}_\pm, \qquad [\hat{L}_+, \hat{L}_-] = 2\hat{L}_0. \tag{106}$$

Note that $\hat{L}_0|\Upsilon\rangle = \frac{1}{2}\lambda(1-\lambda)|\Upsilon\rangle$. Moreover, $\hat{L}_+$ annihilates $|\Upsilon\rangle$, whereas $\hat{L}_-$ acts as

$$\hat{L}_-|\Upsilon\rangle = \left((1-\lambda)\hat{f}_1^+(\hat{f}_0^+)^\dagger + \lambda(\hat{f}_{-1}^+)^\dagger\hat{f}_0^+\right)|\Upsilon\rangle. \tag{107}$$

Proceeding further, we build a tower of states:

$$\mathcal{V} = \{|\Upsilon\rangle, \hat{L}_-|\Upsilon\rangle, \hat{L}_-^2|\Upsilon\rangle, \dots\} . \tag{108}$$

This is a discrete series representation of weight $\Delta = \lambda(1-\lambda)/2$, with $\hat{L}_-$ acting as the raising operator. Computing the Casimir gives:

$$\mathfrak{C}_2|\Upsilon\rangle = \left(-\hat{L}_0^2 + \frac{1}{2}(\hat{L}_+\hat{L}_- + \hat{L}_-\hat{L}_+)\right)|\Upsilon\rangle = \Delta(1-\Delta)|\Upsilon\rangle . \tag{109}$$

Note that $\Delta$ is generally not an integer. Thus, $\mathcal{V}$ is a representation of the universal cover of $SL(2,\mathbb{R})$, with centre character $\mu = \pm\Delta$.

What about the transformation properties of the operators $\hat{f}_n^+$ and $(\hat{f}_n^+)^\dagger$? By commuting them with the $\hat{L}_a$, we find,

$$[\hat{L}_\pm, \hat{f}_{-n}^+] = (\pm(\lambda-1)-n)\hat{f}_{-n\mp 1}^+ , \qquad [\hat{L}_0, \hat{f}_{-n}^+] = -n\hat{f}_{-n}^+ , \tag{110}$$

and similarly for $(\hat{f}_n^+)^\dagger$. Thus, the $\hat{f}_n^+$ and $(\hat{f}_n^+)^\dagger$ transform in the principal series representation. One can construct a principal series multiplet as: $\mathcal{V}_p = \{\hat{f}_n^+ | n \in \mathbb{Z}\}$ with weight $\lambda = 1/2 + iq$, or $\mathcal{V}_p^* = \{(\hat{f}_n^+)^\dagger | n \in \mathbb{Z}\}$ with shadow weight $\lambda^* = 1/2 - iq$.

*Vanishing q*

The presence of an infinite tower in $\mathcal{V}_p$, extending along positive and negative $n$, is related to a non-standard choice for the hermiticity condition for the $\hat{f}_n^+$. This choice is natural for non-vanishing $q$. When $q$ vanishes the quantum fields no longer sense the background electric field. In this case standard Dirichlet boundary conditions can be imposed, under which one obtains the usual quantisation condition for the frequency. Moreover, for vanishing $q$ one inherits the standard hermiticity condition $\hat{f}_{-n}^+ = (\hat{f}_n^+)^\dagger$. This forbids the principal series representation from appearing in the single-particle Hilbert space. In turn, this allows the construction of an $SL(2,\mathbb{R})$ invariant state on top of which we can build a bounded spectrum. Finally, the symmetry group of quantum fields that are both neutral and massless becomes the full Virasoro group, which again invalidates the principal series as a unitary irreducible representation [46]. Thus, one should take caution in taking the $q \to 0$ limit.

## 6.2 $|\Omega\rangle$ as a limit of $|\Upsilon\rangle$

The state $|\Upsilon\rangle$ van be viewed as a filled fermi sea about $|\Omega\rangle$. The Hilbert space built on top of the $|\Upsilon\rangle$ has an $\hat{L}_0$ eigenspectrum which is bounded from below. Here, we discuss the construction of a state in the $|\Upsilon\rangle$ Hilbert space that approximates $|\Omega\rangle$. Define the following

$$|\Omega_N\rangle \equiv \prod_{n=0}^{N} \hat{f}_n^+ |\Upsilon\rangle , \tag{111}$$

with $N$ being some large positive integer. We note that,

$$\langle\Omega_N| \hat{f}_n^+ (\hat{f}_m^+)^\dagger |\Omega_N\rangle = \begin{cases} \delta_{nm} & n = m \leq N , \\ 0 & \text{otherwise} , \end{cases} \tag{112}$$

from which it follows that

$$\langle\Omega_N| \hat{\mathcal{O}}_\lambda(0)\hat{\mathcal{P}}_\lambda(\tau)|\Omega_N\rangle = \sum_{n\leq N} \frac{\Gamma(\lambda+n)}{\Gamma(1-\lambda+n)} e^{-in\tau} . \tag{113}$$

The above sum can be evaluated in terms of a hypergeometric function. Explicitly,

$$\langle \Omega_N | \hat{\mathcal{O}}_\lambda(0) \hat{\mathcal{P}}_\lambda(\tau) | \Omega_N \rangle = \frac{\Gamma(2\lambda)\cos(\pi\lambda)}{2^{2\lambda-1}} \left( \sin^{-2}\tau/2 \right)^{1/2+iq} - \mathcal{F}_N(\tau) \,, \qquad (114)$$

where

$$\mathcal{F}_N(\tau) \equiv e^{-i(N+1)\tau} \frac{\Gamma\left(\frac{3}{2} + iq + N\right)}{\Gamma\left(\frac{3}{2} - iq + N\right)} \,_2F_1\left(1, \frac{3}{2} + iq + N; \frac{3}{2} - iq + N; e^{-i\tau}\right) \,. \qquad (115)$$

Taking the large $N$ limit of $\mathcal{F}_N(\tau)$, we find

$$\lim_{N\to\infty} \mathcal{F}_N(\tau) = \frac{N^{2iq}}{1 - e^{-i\tau}} e^{-iN\tau} \,, \qquad (116)$$

provided $\tau$ is slightly away from $\tau = 2\pi n$ with $n \in \mathbb{Z}$. It is now clear from (116) that the correction to the two-point function (102) is a highly oscillatory function in $\tau$. Consequently, it is convenient to consider the two-point function of slightly smeared boundary operators such as

$$\tilde{\mathcal{P}}_\lambda(\tau_0) = \frac{1}{\sqrt{2\pi\epsilon}} \int_{\mathcal{I}} d\tau\, e^{-\frac{(\tau-\tau_0)^2}{2\epsilon}} \hat{\mathcal{P}}_\lambda(\tau) \,, \qquad (117)$$

where $\epsilon$ is a small positive number. The integral is over a suitably chosen interval $\mathcal{I}$ centred around $\tau_0$, such that $\mathcal{I}$ is contained within the (open) interval $(0, 2\pi)$. We similarly define a smeared operator $\tilde{\mathcal{O}}(\tau)$. It then follows that

$$\lim_{N\to\infty} \langle \Omega_N | \tilde{\mathcal{O}}_\lambda(0) \tilde{\mathcal{P}}_\lambda(\tau) | \Omega_N \rangle = \langle \Omega | \hat{\mathcal{O}}_\lambda(0) \hat{\mathcal{P}}_\lambda(\tau) | \Omega \rangle \,, \qquad (118)$$

up to exponentially small corrections in $N$. Here, we have taken the limit $N \to \infty$ while keeping the product $N\epsilon$ of order one. In a similar way we can recover other observables computed in the $|\Omega\rangle$ state.

# 7 Outlook

Motivated by the near horizon geometry of the extreme Kerr black hole, we have explored quantum field theory on a fixed $\text{AdS}_2$ geometry endowed with a constant background electric field. In doing so, we uncovered the presence of various irreducible representations of the $SL(2,\mathbb{R})$ symmetry group. Of these, the principal series representation, long known to be unitary, was found to describe states carrying flux across the $\text{AdS}_2$ boundary. The presence of incoming and outgoing flux at the $\text{AdS}_2$ boundary suggests the construction of a novel observable more akin to the flat space $S$-matrix. We propose such an observable for $\text{AdS}_2$, the $\mathcal{S}$-matrix. In order to construct it we append global $\text{AdS}_2$ to an auxiliary two-dimensional Minkowski spacetime. The $\mathcal{S}$-matrix is then defined as the overlap of asymptotic Fock states at the past and future null infinities.

Though the representation theory of $SL(2,\mathbb{R})$ ensures that states furnishing the principal series representation can be unitarily accommodated in the Hilbert space of an $SL(2,\mathbb{R})$ invariant theory, it does not ensure the boundedness of the Hamiltonian operator. Indeed, for charged scalar fields the Hamiltonian operator was shown to be unbounded. On the other hand, charged fermionic fields allow for a state $|\Upsilon\rangle$ on top of which a positive definite spectrum can be constructed. From the perspective of the Kerr black hole, the unboundedness of the Hamiltonian is intimately related to the absence of a Killing vector which is everywhere timelike outside the horizon. Thus, there may be a sense in which boundedness of the Hamiltonian is a condition that should be relaxed, although one must still replace it with a reasonable

physical principle. This principle might be related to the existence of a well-defined $\mathcal{S}$-matrix.

There are several natural directions left to explore.

*Perturbative $\mathcal{S}$-matrix*

It will be interesting to study the effect of small interactions on the $\mathcal{S}$-matrix and map several of the established properties of the flat space $S$-matrix to ones for the AdS$_2$ $\mathcal{S}$-matrix. Moreover, one could investigate whether interactions might cause a transmutation between principal series and highest weight representations.

*Backreaction*

One would also like to understand the physics in the case where we allow the background fields to become dynamical. As a first approximation, one might do so for a purely two-dimensional Maxwell-dilaton-gravity model coupled to the charged matter fields. Perhaps the simplest such toy model would be

$$S = \int d^2 x \sqrt{-g} \left[ \frac{\Phi}{16\pi G} \left( R + \frac{2}{\ell^2} \right) - \frac{1}{4} F_{\mu\nu} F^{\mu\nu} - i \bar{\Psi} \gamma^\mu \mathcal{D}_\mu^A \Psi \right] . \tag{119}$$

In the absence of matter, the above action admits an AdS$_2$ vacuum with a constant background electric field [15, 58–60]. As noted in the main text, the massless, charged free fermion is integrable in a general background with general gauge field $A_\mu(\tau, \rho)$. This implies that the action (119) is exactly solvable at the classical level. We will return to this model in the future.

*Microscopic construction?*

It would be interesting to understand the principal series representations from the perspective of recent developments in the AdS$_2$/CFT$_1$ correspondence [2–8]. A concrete model related to our discussion would be an SYK type model with a $U(1)$ global symmetry. The chemical potential $\alpha$ for the $U(1)$ should be exactly marginal at large $N$. The presence of a marginal deformation corresponds to the fact that the background electric field preserves the $SL(2,\mathbb{R})$ symmetry. Moreover, the imaginary part of the conformal weights should vary with $\alpha$. So far, most variations of the SYK model contain conformal operators with real weights. However there are some notable models that might be relevant to our discussion. The models in [9] have a $U(1)$ whose charge operator is an exactly marginal deformation. The fermionic operators in these models have weight $\Delta_\psi = 1/2$, which can be viewed as a limit of the principal series with vanishing imaginary part. The model considered in [61] was shown to have a scalar operator with weight $\Delta = 1/2 + is$ with $s \in \mathbb{R}$ given approximately by $s \approx 1.525$. The authors suggest that this mode implies an instability (see also [62]). Nevertheless, given that the conformal weight has the principal series form one is tempted to search for similar models, containing weights of the form $\Delta = 1/2 + is$ as well as a $U(1)$ global symmetry. Given that superradiance is the rotational analogue of Hawking radiation, the constructions of such microscopic models may pave the way toward a microscopic realisation of Hawking radiation itself.

It is also interesting to note that the $|\Upsilon\rangle$ vacuum exhibits an interesting spontaneous symmetry breaking pattern. Holographically, one would expect a theory which at large $N$ has an approximately $SL(2,\mathbb{R})$ invariant state describing the AdS$_2$ background. We can view the $|\Upsilon\rangle$ vacuum as a small $SL(2,\mathbb{R})$ breaking contribution to the large $N$ state, which transforms in a highest-weight representation. This symmetry breaking pattern is similar to the one studied in the original conformal quantum mechanical models [1] whose ground state transforms in highest weight representation.

*Conformal bootstrap and the principal series*

Usually, in the conformal bootstrap for (0+1)-dimensional conformal field theories, only highest weight representations of $SL(2,\mathbb{R})$ are considered [63–65]. As a result only these representations appear in the crossing equation. It would be interesting to explore the consequences of having principal series representations as unitary representations at the level of the physical Hilbert space. For instance, how is the state-operator correspondence modified, and how do (0+1)-dimensional conformal field theories with operators transforming in the principal series representations emerge from the bootstrap equations?

Furthermore, although the Lorentzian and Euclidean conformal groups differ in higher dimensions, the Lorentzian conformal group $SO(d-1,2)$ with $d > 2$ in fact admits *unitary* irreducible principal series representations labeled by two continuous parameters as well as a discrete set of rotational labels [66]. These are often discarded on the basis of the boundedness of a judiciously chosen Hamiltonian [67,68]. Perhaps one should reassess their physical relevance in light of their apearence in the study of quantum fields in AdS$_2$.

*An $\mathcal{S}$-matrix for dS$_2$?*

It is also interesting to try to construct an $\mathcal{S}$-matrix for dS$_2$ following the discussion in section 4.2. To do so we imagine taking global dS$_2$ with metric

$$\frac{ds^2}{\ell^2} = -d\tau^2 + \cosh^2\tau \, d\varphi^2 \, . \tag{120}$$

Usually the $\varphi$-direction is compact, but the above geometry remains smooth when we unwrap the $\varphi$-circle [69], with symmetry group given by the universal cover of $SL(2,\mathbb{R})$. We can then consider appending an asymptotically flat region in the far past and future where we construct asymptotic scattering states. It would be interesting to understand the unwieldy infrared features of light fields in dS$_2$ from this perspective. Some other related approaches are discussed in [70,71].

# Acknowledgements

We gratefully acknowledge Tarek Anous, Frederik Denef, Chris Herzog, Igor Klebanov, Finn Larsen, Gui Pimentel, Andrea Puhm, Koenraad Schalm, Edgar Shaghoulian, and Guillermo Silva for useful discussions. JK is supported by the Delta ITP consortium, a program of the Netherlands Organisation for Scientific Research (NWO) that is funded by the Dutch Ministry of Education, Culture and Science (OCW). D.A. is funded by a Royal Society University Research Fellowship *"The Atoms of a de Sitter Universe"*. D.A. would like to thank the KITP workshop "Chaos and Order: From strongly correlated systems" supported in part by the National Science Foundation under Grant No. NSF PHY-1748958, where part of this work was completed. D.H. is supported in part by the ERC Starting Grant GENGEOHOL.

# A  Massive fermion in AdS$_2$

Here we consider a massive free fermion in the Poincaré patch of AdS$_2$ with constant background electric field. This problem was also considered in [72] and for notational convenience, we repeat their analysis here. The Lorentzian action is given by

$$S = \int d^2x \sqrt{-g} \, i \left[ \frac{1}{2} \left( \bar{\Psi}\gamma^\mu \overrightarrow{\mathcal{D}}^A_\mu \Psi - \bar{\Psi} \overleftarrow{\mathcal{D}}^A_\mu \gamma^\mu \Psi \right) - \bar{\Psi}(m - i\tilde{m}\gamma)\Psi \right] , \tag{121}$$

with $\bar{\Psi} = \Psi^\dagger \gamma^0$ and $\gamma^0 = i\sigma_1$, $\gamma^1 = \sigma_3$. The gamma matrices obey the usual Clifford algebra $\{\gamma^\mu, \gamma^\nu\} = 2g^{\mu\nu}$ so that $\gamma^\mu = e^\mu_a \gamma^a$ where the sum is over $a = 0, 1$ and $e^\mu_a$ the inverse twei-bein. The gamma matrix $\gamma$ is then $\gamma = \gamma^0 \gamma^1 = -\sigma_2$. The mass $\tilde{m}$ is a time-reversal breaking mass which arises from the separation constant in Dirac equation from the four dimensional perspective. The wave-equation becomes:

$$\left(\gamma^\mu \mathcal{D}^A_\mu - m + i\tilde{m}\gamma\right)\Psi = 0\,,\tag{122}$$

with a covariant derivative $\overrightarrow{\mathcal{D}}^A_\mu$ and $\overleftarrow{\mathcal{D}}^A_\mu$ that reads:

$$\overrightarrow{\mathcal{D}}^A_\mu = \partial_\mu - \frac{i}{2}\omega^{ab}_\mu \Sigma_{ab} - iqA_\mu\,,\tag{123}$$

$$\overleftarrow{\mathcal{D}}^A_\mu = \overleftarrow{\partial}_\mu + \frac{i}{2}\omega^{ab}_\mu \Sigma_{ab} + iqA_\mu\,,\tag{124}$$

with $\Sigma_{ab} = \frac{i}{4}[\gamma_a, \gamma_b]$ whose only non-zero components are $\Sigma_{10} = -\Sigma_{01} = -\frac{i}{2}\sigma_2$. To be more explicit, let us consider the Poincaré patch of AdS$_2$, which can be obtained from the following twei-beins,

$$e^t_{\underline{t}} = e^z_{\underline{z}} = \frac{1}{z}, \quad e^t_{\underline{t}} = e^z_{\underline{z}} = z,\tag{125}$$

where we also wrote down their inverses. Other components are zero. The spin connection $\omega^{ab}_\mu$ can now be computed straightforwardly and yields

$$\omega^{ab}_\mu dx^\mu = -\varepsilon^{ab}\frac{dt}{z}\,,\tag{126}$$

with $\varepsilon^{ab}$ the Levi-Civita tensor in two dimensions, i.e. $\varepsilon^{\underline{tz}} = -\varepsilon^{\underline{zt}} = 1$. The background gauge field again takes the form

$$A = \frac{dt}{z}.\tag{127}$$

We can now assemble all pieces to construct the wave equation for our fermion. Noting that $\gamma^t = z(i\sigma_1)$ and $\gamma^z = z\sigma_3$, it is

$$\left[iz\sigma_1\left(\partial_t - \frac{1}{2z}\sigma_2 - i\frac{q}{z}\right) + \sigma_3 z\partial_z - m + i\tilde{m}\sigma_2\right]\Psi = 0.\tag{128}$$

Massaging this a little bit, we obtain

$$\left[iz\sigma_1\left(\partial_t - i\frac{q}{z}\right) + \sigma_3 z\left(\partial_z - \frac{1}{2z}\right) - m + i\tilde{m}\sigma_2\right]\Psi = 0,\tag{129}$$

from which it is clear that we can get rid of the spin-connection part by rescaling $\Psi$ as $\Psi = z^{1/2}e^{-i\omega t}\psi$. The Dirac equation for $\psi$ becomes

$$(z\partial_z - i\sigma_2(z\omega + q))\psi = (\sigma_3 m - \tilde{m}\sigma_1)\psi.\tag{130}$$

It is useful to write the operator $\psi$ as $\psi = \frac{1}{\sqrt{2}}(1 + i\sigma_1)\chi$ and multiply both sides of (130) by $\frac{1}{\sqrt{2}}(1 - i\sigma_1)$. This yields

$$\begin{pmatrix} z\partial_z \chi_+ + i(z\omega + q)\chi_+ \\ z\partial_z \chi_- - i(z\omega + q)\chi_- \end{pmatrix} = \begin{pmatrix} (im - \tilde{m})\chi_- \\ (-im - \tilde{m})\chi_+ \end{pmatrix},\tag{131}$$

which has solutions

$$\chi = z^{-1/2}\left[\alpha\begin{pmatrix} W_{1/2-iq,\nu}(2i\omega z) \\ (\tilde{m}+im)W_{-1/2-iq,\nu}(2i\omega z) \end{pmatrix} + \beta\begin{pmatrix} (\tilde{m}-im)W_{-1/2+iq,\nu}(-2i\omega z) \\ W_{1/2+iq,\nu}(-2i\omega z) \end{pmatrix}\right],\tag{132}$$

with $\nu = \sqrt{m^2 + \tilde{m}^2 - q^2}$. This solution reduces to the solution considered in the main text upon setting $m = \tilde{m} = 0$. Notice that such a simplification did not happen for the scalar field and is one of the reasons why the fermion allows for a much cleaner description.

# B  Massive quantum fields in dS$_2$

In this appendix we consider a massive scalar field in a fixed dS$_2$ background. We choose the metric:

$$\frac{ds^2}{\ell^2} = \frac{-d\eta^2 + dx^2}{\eta^2} , \qquad \eta \in (-\infty, 0) , \quad x \in \mathbb{R} . \tag{133}$$

The isometry group of dS$_2$ is $SL(2, \mathbb{R})$. The action is given by:

$$S = \frac{1}{2} \int \frac{d\eta dx}{\eta^2} \left( \eta^2 (\partial_\eta \Phi)^2 - \eta^2 (\partial_x \Phi)^2 - \mu^2 \ell^2 \Phi^2 \right) . \tag{134}$$

Going to momentum space, the solutions to the Klein-Gordon equation are given by:

$$\Phi_{\mathbf{k}}(\eta) = \mathcal{N}(-k\eta)^{1/2} J_{i\nu}(-k\eta) , \qquad \nu \equiv \sqrt{\mu^2 \ell^2 - 1/4} , \tag{135}$$

as well as $\Phi_{\mathbf{k}}^*(\eta)$. We focus on the heavy case where $\nu \in \mathbb{R}$. The late time behaviour is given by:

$$\lim_{k\eta \to 0^-} \Phi_{\mathbf{k}}(\eta) = \frac{\mathcal{N}}{2^{i\nu} \Gamma(1 + i\nu)} (-k\eta)^{1/2 + i\nu} (1 + \mathcal{O}(\eta)) . \tag{136}$$

Using the standard Klein-Gordon norm, we can fix the normalisation to:

$$|\mathcal{N}|^2 = \frac{\pi^2}{k \sinh \pi \nu} . \tag{137}$$

Upon quantising the field, we introduce a complex operator $\hat{\alpha}_{\mathbf{k}}$ such that:

$$\hat{\Phi}(\eta, \mathbf{x}) = \int_{\mathbb{R}} \frac{d\mathbf{k}}{2\pi} \left( \hat{\alpha}_{\mathbf{k}} \Phi_{\mathbf{k}}(\eta) e^{i\mathbf{k}\cdot\mathbf{x}} + h.c. \right) , \tag{138}$$

such that $[\hat{\alpha}_{\mathbf{k}}, \hat{\alpha}_{\mathbf{k}'}^\dagger] = \delta_{\mathbf{k}\mathbf{k}'}$. We can define a position space complex operator:

$$\hat{\alpha}(\mathbf{x}) = \int_{\mathbb{R}} \frac{d\mathbf{k}}{2\pi} e^{i\mathbf{k}\cdot\mathbf{x}} k^{i\nu} \hat{\alpha}_{\mathbf{k}} . \tag{139}$$

This transforms as an $SL(2, \mathbb{R})$ conformal operator of weight $\Delta = 1/2 + i\nu$. Similarly, $\hat{\alpha}^\dagger(\mathbf{x})$ will transform with the shadow weight $\bar{\Delta} = 1/2 - i\nu$. The operator algebra now becomes:

$$[\hat{\alpha}(\mathbf{x}), \hat{\alpha}^\dagger(\mathbf{y})] = \delta(\mathbf{x} - \mathbf{y}) . \tag{140}$$

In the far past, where $k\eta \to -\infty$, $\Phi_{\mathbf{k}}(\eta)$ is a linear combination of positive and negative frequency modes. However, one can find a different basis of solutions that are purely positive/negative frequency, such that standard Minkowski quantum fields are recovered deep inside the dS horizon. Explicitly,

$$\lim_{k\eta \to -\infty} \frac{1}{\sqrt{e^{2\pi\nu} - 1}} \left( \Phi_{\mathbf{k}}(\eta) - e^{\pi\nu} \Phi_{\mathbf{k}}^*(\eta) \right) = \sqrt{\frac{\pi}{k}} e^{ik\eta} . \tag{141}$$

For the Minkoswki basis, we can introduce creation and annihilation operators $\hat{a}_{\mathbf{k}}$ and $\hat{a}_{\mathbf{k}}^\dagger$. These will be linear combinations of the $\hat{\alpha}_{\mathbf{k}}$ and $\hat{\alpha}_{\mathbf{k}}^\dagger$, for instance:

$$\hat{a}_{\mathbf{k}} = \frac{\left( \hat{a}_{\mathbf{k}} - e^{\pi\nu} \hat{a}_{\mathbf{k}}^\dagger \right)}{\sqrt{e^{2\pi\nu} - 1}} , \qquad \hat{a}_{\mathbf{k}}^\dagger = \frac{\left( \hat{a}_{\mathbf{k}}^\dagger - e^{\pi\nu} \hat{a}_{\mathbf{k}} \right)}{\sqrt{e^{2\pi\nu} - 1}} . \tag{142}$$

The $SL(2,\mathbb{R})$ invariant Bunch-Davies vacuum $|0\rangle$ is annihilated by $\hat{a}_{\mathbf{k}}$, but it is *not* annihilated by either $\hat{\alpha}_{\mathbf{k}}$ or $\hat{\alpha}_{\mathbf{k}}^{\dagger}$. Rather, it is a highly populated coherent state in terms of the Fock space constructed by the vacuum annihilated by $\hat{\alpha}_{\mathbf{k}}$. The late-time two-point function in the Bunch-Davies state is given by:

$$\lim_{k\eta\to0^-}\langle0|\hat{\Phi}_{\mathbf{k}}(\eta)\hat{\Phi}_{\mathbf{k}'}(\eta)|0\rangle = -\frac{\eta}{2}\left|2^{i\nu}e^{\frac{\pi\nu}{2}}\Gamma(i\nu)(-k\eta)^{-i\nu}+2^{-i\nu}e^{-\frac{\pi\nu}{2}}\Gamma(-i\nu)(-k\eta)^{i\nu}\right|^2\delta_{\mathbf{kk}'}. \tag{143}$$

As a function of $k$, the above is not an $SL(2,\mathbb{R})$ invariant correlation function.[10] Rather, it is a linear combination of invariant correlators with complex weights. On the other hand, it follows from conformal symmetry that in any $SL(2,\mathbb{R})$ invariant state:

$$\langle\hat{\alpha}(\mathbf{x})\hat{\alpha}(\mathbf{y})\rangle = \frac{c_\alpha}{|\mathbf{x}-\mathbf{y}|^{1+2i\nu}}, \qquad c_\alpha\in\mathbb{C}, \qquad \mathbf{x}\neq\mathbf{y}. \tag{144}$$

That the correlator yields a complex answer is consistent with the operator being complex rather than Hermitean. Hence we see how dS$_2$ unitarily realises the principal series representations of $SL(2,\mathbb{R})$. We should also note that there are many $SL(2,\mathbb{R})$ invariant states we can construct [73]. For instance, given the Bunch-Davies vacuum $|0\rangle$, we can act with the non-local composite operator $\hat{\mathcal{O}}=\int d\mathbf{x}\,\hat{\alpha}(\mathbf{x})\hat{\alpha}^{\dagger}(\mathbf{x})$ to construct another conformally invariant state. The Bunch-Davies vacuum has the virtue of being a state with no particles in the Minkowski Fock basis.

## B.1 The case $\nu=0$

It is of interest to understand the case $\nu=0$ as this is a special point separating two different representations. At $\nu=0$, our general solution becomes:

$$\Phi_{\mathbf{k}}(\eta) = \sqrt{-k\eta}\left(\alpha_{\mathbf{k}}J_0(-k\eta)+\beta_{\mathbf{k}}Y_0(-k\eta)\right). \tag{145}$$

The Euclidean modes are given by:

$$\Phi_{\mathbf{k}}^E(\eta) = \frac{1}{2}\sqrt{-\pi\eta}\left(J_0(-k\eta)+iY_0(-k\eta)\right). \tag{146}$$

The late time operators $\hat{\alpha}_{\mathbf{k}}$ and $\hat{\beta}_{\mathbf{k}}$ are now Hermitean, such that:

$$\lim_{\eta\to0}\hat{\Phi}(\eta,\mathbf{x}) = \int_{\mathbb{R}}\frac{d\mathbf{k}}{2\pi}e^{i\mathbf{k}\cdot\mathbf{x}}\left(\chi_{\mathbf{k}}(\eta)\hat{\alpha}_{\mathbf{k}}+\xi_{\mathbf{k}}(\eta)\hat{\beta}_{\mathbf{k}}\right). \tag{147}$$

We have introduced the mode functions $\chi_{\mathbf{k}}(\eta)$ and $\xi_{\mathbf{k}}(\eta)$ which behave as $\sim\sqrt{-\eta}$ and $\sim\sqrt{-\eta}\log(-k\eta)$ at late times. The conformal operator algebra now becomes:

$$[\hat{\alpha}(\mathbf{x}),\hat{\beta}(\mathbf{y})] = i\delta(\mathbf{x}-\mathbf{y}). \tag{148}$$

In an $SL(2,\mathbb{R})$ invariant state $|0\rangle$ which is not annihilated by $\hat{\alpha}(\mathbf{x})$ or $\hat{\beta}(\mathbf{x})$ we have:

$$\langle0|\hat{\alpha}(\mathbf{x})\hat{\alpha}(\mathbf{y})|0\rangle = c_\alpha\delta(\mathbf{x}-\mathbf{y}), \qquad \langle0|\hat{\beta}(\mathbf{x})\hat{\beta}(\mathbf{y})|0\rangle = \frac{c_\beta}{|\mathbf{x}-\mathbf{y}|}. \tag{149}$$

The $\hat{\alpha}(\mathbf{x})$ correlator is somewhat unusual, in that it is ultra-local. Related to this, near $\eta\to-\infty$ the mode $\chi_{\mathbf{k}}(\eta)$ consists of a left and right moving wave with equal amplitude.

---

[10]This is different than the fact that it is de Sitter invariant.

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
