# Peer review of "Charged Quantum Fields in AdS$_2$"

_SciPost Physics, doi:SciPost Phys. 7, 054 (2019)_

## Round 1 · Referee Report · Anonymous (Referee 2) · 2019-9-8

Report

The paper considers the quantization of charged fields in AdS2 in the presence of a background electric field. This problem comes from thinking about the near-horizon geometry of an extremal Kerr black hole. The paper emphasizes the representation-theoretic significance of various states. In particular, states carrying non-trivial flux at the asymptotic boundary of AdS2 are shown to correspond to the principal series representations of SL(2,R). These representations are usually discarded on the basis of their energy being unbounded from below. However, the paper makes a convincing case that they should be considered in this set-up. The paper also defines an interesting new quantity which corresponds to the S-matrix of these states.

The paper is very well-written and provides several new insights into the Kerr/CFT correspondence and quantum field theory in general. I recommend that it can be published in its current form.

---

## Round 1 · Referee Report · Anonymous (Referee 1) · 2019-9-8

Strengths

1- a novel UV completion of AdS2 gravity with an asymptotically flat region, which allows a definition of S-matrix for modes with complex conformal weight 2- it showed that the wave equation of a charged massless fermion on the general background is exactly solvable. 3- the quantization and the definition of vacua is discussed 4- It gives a nice review of SL(2) generators and their representations, and explicitly points out the relation between the principal series representation and the flux carrying modes.

Weaknesses

While the paper is generally well written, it would be better to make proper references to previous work in various places. In particular, the connection between the results in the paper and previous work on superradiance and geometries with SL(2)xU(1) isometries is not discussed.

Report

The authors study quantum field theory on the near horizon throat of extreme Kerr black holes, in the language of AdS2 gravity. In particular, the authors focus on the perturbative modes with complex conformal weight and carrying flux through the Poincare AdS2. Such modes were previously discussed in reference [10] and [15], where the connections with superradiance were observed. This paper provides an interesting new handle on the problem by embedding AdS2 in an asymptotically flat spacetime (4.13), which allows for a definition of the S-matrix, a novel observable for the flux-carrying modes. It is also noticed that charged massless fermions are exactly solvable on the new background (4.13). Furthermore, the quantization of fermions and different vacua are discussed.

To conclude, the paper provides an interesting new approach to the problem of superradiance on extremal Kerr black holes, and contains a collection of interesting observations. I would like to recommend the paper for publication after the authors consider the suggestions in the "requested changes" section.

Requested changes

  1. The discussion in the paper would be strengthened if there is a gravitational theory for which the general background (4.13) is a solution. This might be related to point 2 below. It would be good if the authors could comment on this.

  2. Regarding the backreaction, the Maxwell-dilaton-gravity in [16] seems to be relevant, as it is obtained from dimensional reduction of Einstein gravity on the background eq.(2.1). Besides, a detailed holographic dictionary for yet another Maxwell-dilaton-gravity has been worked out in arXiv 1608.07018. The three models [16], [57] and arXiv 1608.07018, mainly differ in the coupling between the dilaton and the gauge field. As a result, the phase space of these models are different. As such difference might play a role in the discussion of backreaction, it would be good if the authors can comment on these other models as well.

3-The authors defined two vacua. It would be helpful to comment on the connections to the Frolov-Thorne vacuum or the Unruh vacuum for the Kerr black holes.

4-I think the authors should properly cite previous works on superradiance of extremal Kerr black holes. For example, Reference [11], [15] and arXiv 0908.3909.

5-Solutions to classical wave equations for scalars, fermions, photons and gravitons on a finite temperature version of eq.(2.1) have been discussed previously in reference [15] and 0908.3909. It seems that the calculation in Poincare AdS2 (for example section 3.2) is equivalent to the near region calculation of the aforementioned papers, in the strictly extremal limit. It would be good to clarify the relations.

6-I think [11] should be mentioned in the discussion of "Dirac sea and the ergosphere", where connections between Fermi-sea, superradiance and the ergosphere has been discussed for 5d extremal black holes.

7-On page 30, it is mentioned that possible microscopic construction would be an SYK type-model with global U(1) symmetry. A notable model with complex fermions were discussed in reference [5] of the paper and arXiv 1612.00849. It would be helpful if the authors could comment on whether this model might be relevant to the current paper.

---

## Round 1 · Referee Report · Anonymous (Referee 3) · 2019-9-14

Strengths

  1. explicit and thorough analysis of charged bosons and fermions in AdS2
  2. identification of SL(2,R) principal series as wave functions that allow flux leaking through the horizon.
  3. S-matrix observable for the states associated to principal series

Weaknesses

see requested changes

Report

Motivated by the near horizon geometry of an extremal Kerr black hole, the paper conducts a thorough analysis of scalar and fermionic fields charged under a U(1) gauge field in a fixed AdS2 background. The quantum mechanics of the system is naturally organized by the SL(2,R) symmetry of the background. In particular, the authors gave explicit formulae for the symmetry charges and wave functions in relation to SL(2,R) representation theory, where the "non-conventional" principal series has a physical interpretation as describing waves that carry flux through the Poincare horizon of AdS2. The authors then went on to define an S-matrix-like observable for such states. The paper is very concrete and well-written. The computations presented here will be useful for a variety future investigations (e.g. AdS2/CFT1, similar story in dS2), among which the connection to Kerr/CFT is particularly interesting. I recommend this paper for publication after the requested changes are considered.

Requested changes

  1. The authors mentioned in section 2.1 that the rep theory for SL(2,R) and that for its universal cover differ. In fact, for Lorentzian AdS2, the latter is more natural since the killing vectors do not generate close loops (there's no compact U(1) subgroup). The authors should explain/justify why they restrict to SL(2,R) rep theory. (The authors actually consider representations of the universal cover of SL(2,R) later in section 6.1 ...)
  2. The background gauge field in (3.4) is invariant under SL(2,R) only after a gauge transformation. Perhaps this is worth mentioning below (3.4).
  3. In (3.7) and below (3.8), the constant part of $\alpha_\mu$ is ambiguous with the authors' definition. This ambiguity should be fixed by demanding the U(1) and SL(2,R) currents to be orthogonal.
  4. Below (3.15), by "...for large enough values of r, the parameter λ becomes complex..." the authors probably meant " for small enough values of $r$..."
  5. in section 4.2, the authors' procedure of defining the "S-matrix" for principal series involve gluing a flat space region to the boundary of AdS2, which is not a smooth procedure. It would be good to comment on how this observable would depend on regularizations of the gluing.
  6. typo in the last paragraph before section 5.3, "... a particularly explicit treatment in encountered ..." should be "... a particularly explicit treatment encountered ..."

---

## Round 2 · Author Response

We thank the referees for their careful assessment of our manuscript and their comments and suggestions. Below we will respond to their requested changes.

Report 1 1. We thank the referee for the comment. The discussion in our paper is at the level of QFT on a fixed curved background. As such, we believe it would obscure the presentation to focus on the dynamical gravitational theory in main body of the text. The only place where we mention this briefly is in the discussion. We address this in point 2 below. 2. We appreciate the referee’s suggestion. We will include citations [16], [57], 1608.07018 in our discussion section. The level at which we discuss backreaction in the discussion section is schematic. We believe a detailed analysis of such models deserves a study in and of its own right. As such, we would prefer to defer this to future work. 3. We thank the referee for bringing up this point. The two fermionic vacua are defined on the global SL(2,R) invariant geometry and are shown to preserve the SL(2,R) symmetry. Thus, our fermionic vacua are not directly connected to vacua adapted to the presence of horizons. In particular they correspond to neither the Frolov-Thorne vacuum (which requires the introduction of a mirror), or the Unruh vacuum (whose stress tensor diverges at the past horizon). 4. As requested by the referee, we have added the necessary citations (hep-th/9905099, [11], 0906.1819, [15], and 0908.3909) at the appropriate point in the main body of the text (i.e. page 3 before “Another motivation....”) 5. These hypergeometric function have appeared repeatedly in the AdS/CFT literature, in the discussion of NHEK, and dS geometries for example. They are well known in the community and it would be impossible to compile all the relevant papers that have looked into these solutions. Having said that, we can confirm that it is indeed the case that the computation in the near region of the mentioned papers is equivalent to ours. 6. We thank the referee for mentioning this. We have added reference to [11] to the `Dirac Sea and ergosphere’ discussion right above section 6. 7. Reference [5] is indeed cited in the discussion of the microscopic models and we mention it presents a real weight for the fermion operator. Therefore, it does not obviously apply to our discussion. Other more promising models are discussed in this section.

Report 3 1. As the referee points out, it is the universal cover that we are mostly interested in. Upon consideration of the referee’s remark, and to make this more manifest, we have changed the wording in section 2.1 from “If one is interested...” to “We will mostly be interested...”. 2. We thank the referee for their remark. We added a comment below 3.4 where we state that the field strength is indeed SL(2,R) invariant. 3. We have added a footnote incorporating the referee’s comment. 4. We are grateful to the referee for picking this up. We have made the requested correction. Also we have changed r to calligraphic R, as we have used r for a coordinate elsewhere. 5. We appreciate the referee’s remark. In general the gluing can be made smooth as discussed near equation (4.13). For the specific case of the model in section 5, we actually address this issue in the “Closing the open quantum system” subsection. 6. We thank the referee for pointing out the typo. It has been fixed in the current version.

You are currently on this page

Resubmission 1906.00924v2 on 9 October 2019

---

## Editorial Decision

published